# Cryptographic Hardness of Score Estimation

**Min Jae Song**
Paul G. Allen School of Computer Science and Engineering
University of Washington
mjsong32@cs.washington.edu

## Abstract

We show that $L^2$-accurate score estimation, in the absence of strong assumptions on the data distribution, is computationally hard even when sample complexity is polynomial in the relevant problem parameters. Our reduction builds on the result of Chen et al. (ICLR 2023), who showed that the problem of generating samples from an unknown data distribution reduces to $L^2$-accurate score estimation. Our hard-to-estimate distributions are the "Gaussian pancakes" distributions, originally due to Diakonikolas et al. (FOCS 2017), which have been shown to be computationally indistinguishable from the standard Gaussian under widely believed hardness assumptions from lattice-based cryptography (Bruna et al., STOC 2021; Gupte et al., FOCS 2022).

## 1 Introduction

Diffusion models [70, 72, 42, 73] have firmly established themselves as a powerful approach to generative modeling, serving as the foundation for leading image generation models such as DALL-E 2 [62], Imagen [66], and Stable Diffusion [65]. A diffusion model consists of a pair of forward and reverse processes. In the forward process, noise drawn from a standard distribution, such as the standard Gaussian, is sequentially applied to data samples, leading its distribution to a pure noise distribution in the limit. The reverse process, as the name suggests, reverses the noising process and takes the pure noise distribution "backward in time" to the original data distribution, thereby allowing us to generate new samples from the data distribution. A key element in implementing the reverse process is the *score function* of the data distribution, which is the gradient of its log density. Since the data distribution is typically unknown, the score function must be learned from samples [44, 78, 72].

Recent advances in the theory of diffusion models have revealed that the task of sampling, in fact, reduces to score estimation under minimal assumptions on the data distribution [9, 23, 53, 61, 22, 17, 51, 6]. In particular, Chen et al. [17] have shown that $L^2$-accurate score estimates along the forward process are sufficient for efficient sampling. Thus, assuming access to an oracle for $L^2$-accurate score estimation, one can efficiently sample from essentially any data distribution. However, this leaves open the question of whether score estimation oracles themselves can be implemented efficiently, in terms of both required sample size and computation, for interesting classes of distributions.

We show that $L^2$-accurate score estimation, in the absence of strong assumptions on the data distribution, is computationally hard, even when sample complexity is polynomial in the relevant problem parameters. This establishes a *statistical-to-computational gap* for $L^2$-accurate score estimation, which refers to an intrinsic gap between what is statistically achievable and computationally feasible. Our hard-to-estimate distributions are the "Gaussian pancakes" distributions, which previous works [31, 13, 39] have shown are computationally indistinguishable from the standard Gaussian under plausible and widely believed hardness assumptions. In fact, "breaking" the hardness of Gaussian pancakes, by means of an efficient detection or estimation algorithm, has profound implications for lattice-based cryptography, which is central to the post-quantum cryptography standardization led by the National Institute of Standards and Technology (NIST) [58]. Building on the sampling-to-

38th Conference on Neural Information Processing Systems (NeurIPS 2024).

score estimation reduction by Chen et al. [17], we show that computationally efficient $L^2$-accurate score estimation for Gaussian pancakes implies an efficient algorithm for distinguishing Gaussian pancakes from the standard Gaussian. Thus, while sampling may ultimately reduce to $L^2$-accurate score estimation under minimal assumptions on the data distribution, score estimation itself requires stronger assumptions on the data distribution for computational feasibility. It is worth noting that the presence of statistical-to-computational gaps in $L^2$-accurate score estimation was anticipated by Chen et al. [17, Section 1.1], who mentioned it without formal statement or proof.

## 1.1 Main contributions

Our main result is a simple reduction from the Gaussian pancakes problem (i.e., the problem of distinguishing Gaussian pancakes from the standard Gaussian) to $L^2$-accurate score estimation. We show that given oracle access to $L^2$-accurate score estimates along the forward process (Assumption A3), one can compute a test statistic that distinguishes, with non-trivial success probability, whether the given score estimates belong to a Gaussian pancakes distribution or the standard Gaussian.

A Gaussian pancakes distribution $P_{\boldsymbol{u}}$ with secret direction $\boldsymbol{u} \in \mathbb{S}^{d-1}$ can be viewed as a "backdoored" Gaussian. It is distributed as a (noisy) discrete Gaussian along the direction $\boldsymbol{u}$ and as a standard Gaussian in the remaining $d-1$ directions (see Figure 1).[1] A *class* of Gaussian pancakes $(P_{\boldsymbol{u}})_{\boldsymbol{u} \in \mathbb{S}^{d-1}}$ is parameterized by two parameters, $\gamma$ and $\sigma$, which govern the spacing and thickness of pancakes, respectively. For instance, a Gaussian pancakes distribution $P_{\boldsymbol{u}}$ with spacing $\gamma$ and thickness $\sigma \approx 0$ is essentially supported on the one-dimensional lattice $(1/\gamma)\mathbb{Z}$ along the secret direction $\boldsymbol{u}$. The Gaussian pancakes *problem*, then, is a sequence of hypothesis testing problems indexed by the data dimension $d \in \mathbb{N}$ in which the goal is to distinguish between samples from a Gaussian pancakes distribution (with unknown $\boldsymbol{u}$) and the standard Gaussian distribution $\mathcal{N}(0, I_d)$ with success probability slightly better than random guessing (see Section 2.3 for formal definitions). Thus, our result can be summarized informally as follows.

**Theorem 1.1** (Informal, see Theorem 3.1). *Let $\gamma(d) > 1, \sigma(d) > 0$ be sequences such that $\sigma \geq 1/\mathrm{poly}(d)$ and the corresponding $(\gamma, \sigma)$-Gaussian pancakes distributions $(P_{\boldsymbol{u}})_{\boldsymbol{u} \in \mathbb{S}^{d-1}}$ all satisfy $\mathrm{TV}(P_{\boldsymbol{u}}, \mathcal{N}(0, I_d)) > 1/2$. Then, there exists a polynomial-time randomized algorithm with access to a score estimation oracle of $L^2$-error $O(1/\sqrt{\log d})$ that solves the Gaussian pancakes problem.*

We emphasize that the hardness of estimating score functions of Gaussian pancakes distributions arises solely from hardness of *learning*. The score function of $P_{\boldsymbol{u}}$ is efficiently approximable by function classes commonly used in practice for generative modeling, such as residual networks [40]. In addition, under the scaling $\sigma\gamma = O(1)$, which includes the cryptographically hard regime, the secret parameter $\boldsymbol{u}$ can be estimated upto $L^2$-error $\eta$ via brute-force search over $\mathbb{S}^{d-1}$ with $\mathrm{poly}(d, \gamma, 1/\eta)$ samples (Theorem 4.2). The estimated parameter $\hat{\boldsymbol{u}}$ in turn enables $L^2$-accurate score estimation (see Section 4). Our estimator, based on projection pursuit [35, 43], may be of independent interest.

We also analyze properties of Gaussian pancakes using Banaszczyk's theorems on the Gaussian mass of lattices [3, 74]. This serves two purposes. Firstly, it allows us to verify that Gaussian pancakes distributions readily satisfy the assumptions for the sampling-to-score estimation reduction of Chen et al. [17], namely Lipschitzness of the score functions along the forward process (Assumption A1). This is necessary as the proof of our main theorem crucially relies on the reduction. Secondly, Banaszczyk's theorems provide simple means of analyzing properties of Gaussian pancakes, which are interesting mathematical objects in their own right. While these theorems are standard tools in lattice-based cryptography (see e.g., [74, 1]), they are likely less known outside the community.

## 1.2 Related work

**Theory of diffusion models.** Recent advances in the theoretical study of diffusion models have focused on convergence rates of discretized reverse processes [9, 23, 53, 61, 22, 17, 51, 6]. Of particular relevance to our work is the result of Chen et al. [17] who showed that $L^2$-accurate score estimates are sufficient to guarantee convergence rates that are polynomial in all the relevant problem parameters under minimal assumptions on the data distribution, namely Lipschitz scores throughout the forward process and finite second moment. Prior studies fell short by requiring strong structural assumptions on the data distribution, such as a log-Sobolev inequality [50, 82], assuming

---

[1] An animated visualization of Gaussian pancakes can be found in [12].

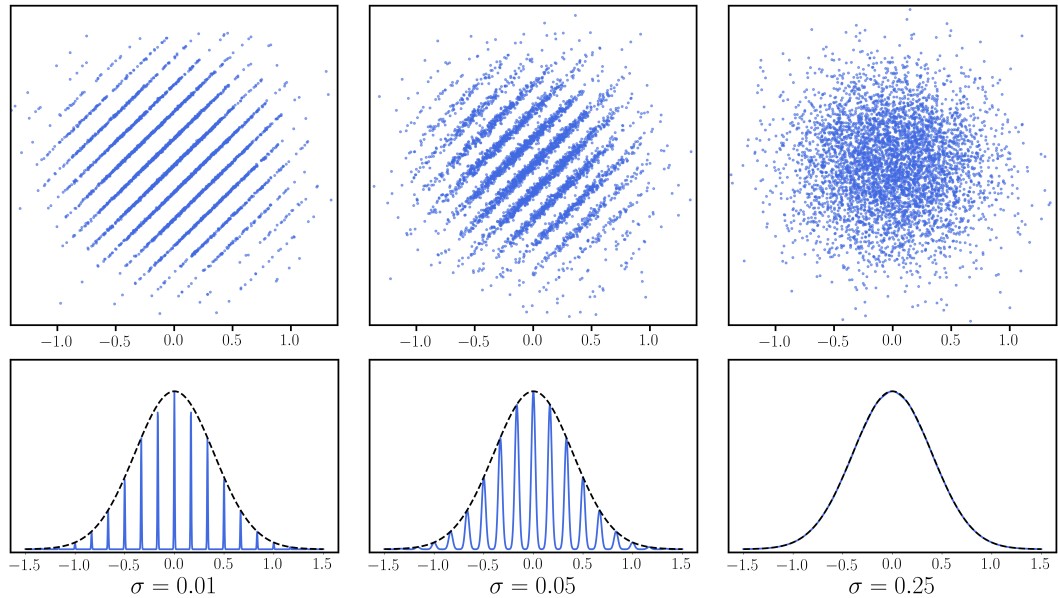

Figure 1: Top: Scatter plot of 2D Gaussian pancakes $P_{\boldsymbol{u}}$ with secret direction $\boldsymbol{u} = (-1/\sqrt{2}, 1/\sqrt{2})$, spacing $\gamma = 6$, and thickness $\sigma \in \{0.01, 0.05, 0.25\}$. Bottom: Re-scaled probability densities of Gaussian pancakes (blue) for each $\sigma \in \{0.01, 0.05, 0.25\}$ and the standard Gaussian (black) along $\boldsymbol{u}$. For fixed $\gamma$, the pancakes "blur into each other" as $\sigma$ increases.

$L^{\infty}$-accurate score estimates [23], or providing convergence rates that are exponential in the problem parameters [9, 23, 22]. We note that recent works have made the "minimal" assumptions of Chen et al. [17] even more minimal by considering early stopped reverse processes and dropping the Lipschitz score assumption [15, 6]. We refer to the book draft of Chewi [19] for more background.

**Gaussian pancakes.** The Gaussian pancakes problem stands out among problems exhibiting statistical-to-computational gaps due to its versatility and strong hardness guarantees. Initially introduced as hard-to-learn Gaussian mixtures in the work of Diakonikolas et al. [31], which established their SQ hardness, Gaussian pancakes have been extensively utilized in establishing SQ lower bounds for various statistical inference problems such as robust Gaussian mean estimation [31] and agnostically learning halfspaces and ReLUs over Gaussian inputs [28]. For further details, we refer to the textbook by Diakonikolas and Kane [29, Chapter 8]. Gaussian pancakes distributions themselves serve as instances of fundamental high-dimensional inference problems such as non-Gaussian component analysis [8] and Wasserstein distance estimation in the spiked transport model [57].

Bruna et al. [13] initiated the exploration of the *cryptographic* hardness of Gaussian pancakes. They showed that assuming hardness of worst-case lattice problems, fundamental to lattice-based cryptography [56, 60], both the Gaussian pancakes and the closely related *continuous* learning with errors (CLWE) problems are hard. Follow-up work by Gupte et al. [39] showed that the learning with errors (LWE) problem [63], a versatile problem which lies at the heart of numerous lattice-based cryptographic constructions, reduces to the Gaussian pancakes as well. These cryptographic hardness results have sparked a wave of recent works showcasing various applications of this newly discovered property of Gaussian pancakes. Notable examples include planting undetectable backdoors in machine learning models [38], novel public-key encryption schemes based on Gaussian pancakes [10], and cryptographic hardness of agnostically learning halfspaces [26, 75].

For additional related work on score estimation and statistical-to-computational gaps, see Section A.

### 1.3 Future directions

Our work brings together the latest advances from the theory of diffusion models and computational complexity of statistical inference. This intersection provides fertile ground for future research, some avenues of which we outline below.

**Stronger data assumptions for efficient score estimation.** Our result shows that for any class of distributions that encompasses hard Gaussian pancakes, computationally efficient $L^2$-accurate score estimation is impossible. This implies that stronger assumptions on the data, which exclude Gaussian pancakes, are necessary for efficient score estimation. Finding assumptions that exclude hard instances while being able to capture realistic models of data is an interesting open problem.

**Weaker criteria for evaluating sample quality.** In the context of learning Gaussian pancakes, if the goal of the sampling algorithm were to merely fool computationally bounded testers that lack knowledge of the secret $u \in \mathbb{S}^{d-1}$, then it could simply generate standard Gaussian samples and fool any polynomial-time test. Thus, sampling is strictly easier than $L^2$-accurate score estimation if the criteria for evaluating sample quality is less stringent. This suggests exploring sampling under different evaluation criteria, such as "discriminators" with bounded compute or memory. For example, Christ et al. [20] have used weaker notions of sample quality in the context of watermarking large language models (LLMs). More precisely, they used the notion of computational indistinguishability to guarantee quality of the watermarked model relative to the original model. Exploring potential connections to the literature on leakage simulation [45, 18, 76] and outcome indistinguishability [41, 32, 33] is also an interesting future direction, as these areas have addressed related questions for distributions on finite sets.

**Extracting "knowledge" from sampling algorithms.** A key difficulty in directly reducing the Gaussian pancakes problem to sampling is that the Gaussian pancakes problem is hard for polynomial-time distinguishers, even with access to *exact* sampling oracles (see Section 2.3 for more details).[2] Thus, any procedure utilizing the learned sampler in a black box manner cannot solve the Gaussian pancakes problem. This is puzzling since for the algorithm to have "learned" to generate samples from the given distribution, it *ought to* possess some non-trivial information about it (e.g., leak information about the secret parameter $u$)!

This raises the question: How much "knowledge" can we extract with *white box* access to the sampling algorithm? Under reasonable structural assumptions on the sampling algorithm, can we extract privileged information about the data distribution it simulates, beyond what is obtainable solely via sample access? Our work provides one such example. White box access to a diffusion model gives access to its score estimates. These score estimates, which enable efficient solutions to the Gaussian pancakes problem, constitute privileged knowledge that cannot be learned efficiently even with unlimited access to bona fide samples.

## 2 Preliminaries

**Notation.** We denote by $(a_t)$ the sequence $a_1, a_2, a_3, \ldots$ indexed by $t \in \mathbb{N}$. When there is no natural ordering on the index set (e.g., $(a_u)_{u \in \mathbb{S}^{d-1}}$), we interpret it as a set. We write $a \lesssim b$ or $a = O(b)$ to mean that $a \leq Cb$ for some universal constant $C > 0$. The notation $a \gtrsim b$ and $a = \Omega(b)$ are defined analogously. We write $a \asymp b$ or $a = \Theta(b)$ to mean that $a \lesssim b$ and $a \gtrsim b$ both hold. We write $\wedge, \vee$ to mean logical AND and OR, respectively. We also write $a \wedge b$ and $a \vee b$ to mean $\min(a, b)$ and $\max(a, b)$, respectively. We denote by $Q_d$ the "standard" Gaussian $\mathcal{N}(0, 1/(2\pi)I_d)$ (see Remark 2.3). We omit the subscript $d$ when it is clear from context.

### 2.1 Background on denoising diffusion probabilistic modeling (DDPM)

We give a brief exposition on *denoising diffusion probabilistic models* (DDPM) [42], a specific type of diffusion model, since the main reduction of Chen et al. [17] pertains to DDPMs. This section closely follows [17, Section 2.1] and [80, Section 3]

Let $D$ be the target distribution defined on $\mathbb{R}^d$. In DDPMs, we begin with the **forward process**, an Ornstein-Uhlenbeck (OU) process that converges towards $Q = \mathcal{N}(0, (1/2\pi)I_d)$, which is described by the following stochastic differential equation (SDE). Note that the constant $1/\sqrt{\pi}$ in front of $dW_t$ in Eq.(1) is non-standard.[3] See Remark 2.3 for an explanation of our unconventional choice of

---

[2]Note, however, that algorithms that run in time $T$ can query the sampling oracle at most $T$ times, so the running time imposes a limit on the number of samples the algorithm can see.

[3]The usual choice is $\sqrt{2}$, for which the stationary distribution is $\mathcal{N}(0, I_d)$.

variance for the resulting stationary distribution $Q$.

$$dX_t = -X_t dt + (1/\sqrt{\pi})dW_t , \qquad X_0 \sim D_0 = D , \qquad (1)$$

where $W_t$ is the standard Brownian motion in $\mathbb{R}^d$.

Let $D_t$ be the distribution of $X_t$ along the OU process. It is well-known that for any distribution $D$, $D_t \to Q$ exponentially fast in various divergences and metrics such as the KL divergence [2]. In this work, we only consider Gaussian pancakes $(P_{\boldsymbol{u}})$ and the standard Gaussian $Q$. If $D = P_{\boldsymbol{u}}$ and $t > 0$, then the distribution $D_t$ is simply another Gaussian pancakes distribution with a larger thickness parameter (see Definition 2.5). Meanwhile, if $D = Q$, then $D_t = Q$ for any $t \geq 0$. We run the OU process until time $T > 0$, and then simulate the **reverse process**, described by the following SDE.

$$dY_t = (Y_t + (1/\pi)\nabla \log D_{T-t}(Y_t))dt + (1/\sqrt{\pi})dW_t , \qquad Y_0 \sim F_0 = D_T . \qquad (2)$$

Here, $\nabla \log D_t$ is called the *score function* of $D_t$. Since the target $D$ is not known, in order to implement the reverse process the score function must be estimated from data samples. Assuming for the moment that we have *exact* scores $(\nabla \log D_t)_{t \in [0,T]}$, if we start the reverse process from $F_0 = D_T$, we have $Y_t \sim F_t = D_{T-t}$ for any $0 \leq t \leq T$, and ultimately $Y_T \sim F_T = D$. Thus, starting from (approximately) pure noise $D_T \approx Q$, the reverse process generates fresh samples from the target distribution $D$. We need to make several approximations to algorithmically implement this reverse process. In particular, we need to approximate $D_T$ by $Q$, discretize the continuous-time SDE, and approximate scores along the (discretized) forward process. Let $h > 0$ be the step size for the SDE discretization and denote $N := T/h$. Given score estimates $(s_{kh})_{k \in [N]}$, the DDPM algorithm performs the following update (see e.g., [67, Chapter 4.3]).

$$\boldsymbol{y}_{k+1} = e^h \boldsymbol{y}_k + (1/\pi)(e^h - 1)s_{(N-k)h}(\boldsymbol{y}_k) + \sqrt{e^{2h} - 1}\boldsymbol{z}_k , \qquad (3)$$

where $\boldsymbol{z}_k \sim Q$ is an independent Gaussian vector. Note that the only dependence of the reverse process on the target distribution $D$ arises through the score estimates $(s_{kh})_{k \in [N]}$.

The result of Chen et al. [17] demonstrates polynomial convergence rates of this process to the target distribution $D$, assuming access to $L^2$-accurate score estimates $(s_{kh})_{k \in [N]}$ and minimal conditions on the target $D$. We refer to Section 3.1 for a formal statement of their assumptions and theorem.

## 2.2 Lattices and discrete Gaussians

**Lattices.** A *lattice* $\mathcal{L} \subset \mathbb{R}^d$ is a discrete additive subgroup of $\mathbb{R}^d$. In this work, we assume all lattices are full rank, i.e., their linear span is $\mathbb{R}^d$. For a $d$-dimensional lattice $\mathcal{L}$, a set of linearly independent vectors $\{\boldsymbol{b}_1, \ldots, \boldsymbol{b}_d\}$ is called a *basis* of $\mathcal{L}$ if $\mathcal{L}$ is generated by the set, i.e., $\mathcal{L} = B\mathbb{Z}^d$ where $B = [\boldsymbol{b}_1, \ldots, \boldsymbol{b}_d]$. The *determinant* of a lattice $\mathcal{L}$ with basis $B$ is defined as $\det(\mathcal{L}) = |\det(B)|$.

The *dual lattice* of a lattice $\mathcal{L}$, denoted by $\mathcal{L}^*$, is defined as

$$\mathcal{L}^* = \{\boldsymbol{y} \in \mathbb{R}^d \mid \langle \boldsymbol{x}, \boldsymbol{y} \rangle \in \mathbb{Z} \text{ for all } \boldsymbol{x} \in \mathcal{L}\} .$$

If $B$ is a basis of $\mathcal{L}$ then $(B^T)^{-1}$ is a basis of $\mathcal{L}^*$; in particular, $\det(\mathcal{L}^*) = \det(\mathcal{L})^{-1}$.

**Fourier analysis.** We define the Fourier transform of a function $f : \mathbb{R}^d \to \mathbb{C}$ by

$$\widehat{f}(\boldsymbol{y}) = \int_{\mathbb{R}^d} f(\boldsymbol{x}) \exp(-2\pi i \langle \boldsymbol{x}, \boldsymbol{y} \rangle) d\boldsymbol{x} .$$

The Poisson summation formula offers a valuable tool for analyzing functions defined on lattices.

**Lemma 2.1** (Poisson summation formula). *For any lattice $\mathcal{L} \subset \mathbb{R}^d$ and any function $f : \mathbb{R}^d \to \mathbb{C}$,*[4]

$$f(\mathcal{L}) = \det(\mathcal{L}^*) \cdot \widehat{f}(\mathcal{L}^*) ,$$

*where we denote $f(S) = \sum_{x \in S} f(x)$ for any set $S$.*

---

[4]To be precise, $f$ must satisfy some niceness conditions; this will always hold in our applications.

**Discrete Gaussians.** A discrete Gaussian is a discrete distribution whose probability mass function is given by the Gaussian function. These distributions are closely related to Gaussian pancakes distributions and their properties will be crucial for our analysis.

**Definition 2.2** (Gaussian function). *We define the Gaussian function $\rho_s : \mathbb{R}^d \to \mathbb{R}$ of width $s > 0$ by*

$$\rho_s(\boldsymbol{x}) := \exp(-\pi\|\boldsymbol{x}\|^2/s^2) \ .$$

*When $s = 1$, we omit the subscript and simply write $\rho$. In addition, for any lattice $\mathcal{L} \subset \mathbb{R}^d$, we denote by $\rho_s(\mathcal{L}) = \sum_{\boldsymbol{x} \in \mathcal{L}} \rho_s(\boldsymbol{x})$ the corresponding Gaussian mass of $\mathcal{L}$.*

**Remark 2.3** (Non-standard choice of "standard" variance). *We refer to $Q_d = \mathcal{N}(0, 1/(2\pi)I_d)$ as the "standard" Gaussian. This is indeed the standard choice in lattice-based cryptography because it simplifies normalization factors that arise from taking Fourier transforms. For instance, it allows us to simply write $\widehat{\rho_s} = s^n\rho_{1/s}$ and $\widehat{\rho} = \rho$ for $s = 1$. To translate these results for the "usual" standard Gaussian, we can simply replace $s$ with $s/\sqrt{2\pi}$ for each occurrence.*

**Definition 2.4** (Discrete Gaussian). *For any lattice $\mathcal{L} \subset \mathbb{R}^d$, parameter $s > 0$, and shift $\boldsymbol{t} \in \mathbb{R}^d$, the discrete Gaussian $D_{\mathcal{L}-\boldsymbol{t},s}$ is a distribution supported on the coset $\mathcal{L} - \boldsymbol{t}$ with probability mass*

$$D_{\mathcal{L}-\boldsymbol{t},s}(\boldsymbol{x}) = \frac{\rho_s(\boldsymbol{x}-\boldsymbol{t})}{\rho_s(\mathcal{L}-\boldsymbol{t})} \ .$$

We denote by $A_\gamma$ the discrete Gaussian of width $s = 1$ supported on the one-dimensional lattice $(1/\gamma)\mathbb{Z}$. The distribution of a Gaussian pancakes distribution $P_{\boldsymbol{u}}$ along the hidden direction is a smoothed discrete Gaussian, which we formalize in the following.

**Definition 2.5** (Smoothed discrete Gaussian). *For any $\gamma > 0$, let $A_\gamma$ be the discrete Gaussian of width 1 on the lattice $(1/\gamma)\mathbb{Z}$. We define the $\sigma$-smoothed discrete Gaussian $A_\gamma^\sigma$ as the distribution of the random variable $y$ induced by the following process.*

$$y = \frac{1}{\sqrt{1+\sigma^2}}(x + \sigma z) \ , \quad \text{where } x \sim A_\gamma \text{ and } z \sim Q \ .$$

*Furthermore, the density of $A_\gamma^\sigma$ is given by*

$$A_\gamma^\sigma(z) = \frac{\sqrt{1+\sigma^2}}{\sigma\rho((1/\gamma)\mathbb{Z})} \sum_{k\in\mathbb{Z}} \rho(k/\gamma)\rho_{\sigma/\sqrt{1+\sigma^2}}\big(z - k/\gamma\sqrt{1+\sigma^2}\big) \ . \tag{4}$$

**Likelihood ratio of smoothed discrete Gaussians.** Let $A_\gamma^\sigma$ be the $\sigma$-smoothed discrete Gaussian on $(1/\gamma)\mathbb{Z}$. Its likelihood ratio $T_\gamma^\sigma$ with respect to the standard Gaussian is given by

$$T_\gamma^\sigma(z) = \frac{\sqrt{1+\sigma^2}}{\sigma\rho((1/\gamma)\mathbb{Z})} \sum_{k\in\mathbb{Z}} \rho_\sigma(z - \sqrt{1+\sigma^2}k/\gamma) \ . \tag{5}$$

When $\gamma$ and $\sigma$ are clear from context, we omit them and simply denote the likelihood ratio by $T$.

## 2.3 Gaussian pancakes

We define Gaussian pancakes distributions using the likelihood ratio $T_\gamma^\sigma = A_\gamma^\sigma/Q$. It is important to note that our parametrization differs from the one used in previous works [13, 39]. We believe our parametrization is more convenient as it elucidates a natural partial ordering on the space of parameters $(\gamma, \sigma)$. In addition, there is an explicit mapping between the two different parametrizations, so computationally hard parameter regimes identified by previous works [13, 39] can readily be translated into setting. See Remark 2.7 for more details.

**Definition 2.6** (Gaussian pancakes). *For any $d \in \mathbb{N}$, spacing and thickness parameters $\gamma, \sigma > 0$, we define the $(\gamma, \sigma)$-Gaussian pancakes distribution $P_{\gamma,\boldsymbol{u}}^\sigma$ with secret direction $\boldsymbol{u} \in \mathbb{S}^{d-1}$ by*

$$P_{\gamma,\boldsymbol{u}}^\sigma(\boldsymbol{x}) := Q(\boldsymbol{x}) \cdot T_\gamma^\sigma(\langle\boldsymbol{x}, \boldsymbol{u}\rangle) \ ,$$

*where $Q = \mathcal{N}(0, (1/2\pi)I_d)$ and $T_\gamma^\sigma$ is the likelihood ratio of $A_\gamma^\sigma$ with respect to $Q$. When parameters $\gamma, \sigma$ are clear from context, we omit them in the notation and simply denote the distribution by $P_{\boldsymbol{u}}$ to avoid clutter.*

**Remark 2.7** (Partial ordering on Gaussian pancakes). *The smoothed discrete Gaussian $A_\gamma^\sigma$ arises in the OU process for the discrete Gaussian $A_\gamma$ at time $t = \log\left(\sqrt{1+\sigma^2}\right)$. Consequently, for any fixed $\gamma > 0$, there exists a natural partial ordering on the family of Gaussian pancakes parametrized by $(\gamma, \sigma)$, given by $(\gamma, \sigma_1) \leq (\gamma, \sigma_2)$ whenever $\sigma_1 \leq \sigma_2$. This ordering arises from the fact that $A_\gamma^{\sigma_1}$ reduces to $A_\gamma^{\sigma_2}$ whenever $\sigma_1 \leq \sigma_2$ via the OU process starting at $t_1 = \log\left(\sqrt{1+\sigma_1^2}\right)$ and run until $t_2 = \log\left(\sqrt{1+\sigma_2^2}\right)$.*

**Definition 2.8** (Advantage). *Let $\mathcal{A} : \mathcal{X} \to \{0, 1\}$ be any decision rule (i.e., distinguisher). For any pair of distributions $(\mathcal{P}, \mathcal{Q})$ on $\mathcal{X}$, we define the* advantage *of $\mathcal{A}$ by*

$$\alpha(\mathcal{A}) := \left| \mathcal{P}[\mathcal{A}(X) = 1] - \mathcal{Q}[\mathcal{A}(X) = 1] \right|.$$

*For a sequence of decision rules $(\mathcal{A}_d)_{d\in\mathbb{N}}$ and distribution pairs $(\mathcal{P}_d, \mathcal{Q}_d)_{d\in\mathbb{N}}$, we say $(\mathcal{A}_d)$ has* non-negligible advantage *with respect to $(\mathcal{P}_d, \mathcal{Q}_d)$ if its advantage sequence $\alpha_d = \alpha(\mathcal{A}_d)$ is a non-negligible function in $d$, i.e., a function in $\Omega(d^{-c})$ for some constant $c > 0$.*

**Definition 2.9** (Computational indistinguishability). *A sequence of distribution pairs $(\mathcal{P}_d, \mathcal{Q}_d)$ is* computationally indistinguishable *if no $\mathrm{poly}(d)$-time computable decision rule achieves non-negligible advantage.*

**Definition 2.10** (Gaussian pancakes problem). *For any sequences $\gamma(d), \sigma(d) > 0$ and $n(d) \in \mathbb{N}$, the $(\gamma, \sigma, n)$-Gaussian pancakes problem is to distinguish $(\mathcal{P}_d, \mathcal{Q}_d)$ with non-negligible advantage, where $\mathcal{P}_d$ is the $n$-sample distribution induced by the following two-stage process: 1) draw $\boldsymbol{u}$ uniformly from $\mathbb{S}^{d-1}$, 2) draw $n$ i.i.d. Gaussian pancakes samples $\boldsymbol{x}_1, \ldots, \boldsymbol{x}_n \sim P_{\boldsymbol{u}}^{\otimes n}$, and $\mathcal{Q}_d = Q_d^{\otimes n}$, i.e., the distribution of $n$ i.i.d. standard Gaussian vectors.*

As will be explained next, the exact number of samples $n$ is irrelevant for most applications due to the cryptographic hardness of the Gaussian pancakes problem. For certain parameter regimes of $(\gamma, \sigma)$, the problem maintains its computational intractability regardless of the sample size $n$.

**Hardness of Gaussian pancakes.** There is an abundance of evidence demonstrating the hardness of the Gaussian pancakes problem. This makes it compelling to directly assume that Gaussian pancakes and the standard Gaussian are *computationally indistinguishable* (see Definition 2.9) for certain parameter regimes of $(\gamma, \sigma)$. Initial results by Bruna et al. [13, Corallary 4.2] showed that the Gaussian pancakes problem is as hard as worst-case lattice problems for any parameter sequence $(\gamma, \sigma)$ satisfying $\gamma \geq 2\sqrt{d}$ and $\sigma \geq 1/\mathrm{poly}(d)$. SQ hardness of the problem has been demonstrated as well [31, 13, 27]. Perhaps surprisingly, the reduction of Bruna et al. shows that even with *unlimited* access to an *exact* sampling oracle, no polynomial-time algorithm $\mathcal{A}$ can achieve non-negligible advantage on the $(\gamma, \sigma)$-Gaussian pancakes problem. This stems from the fact that the running time of $\mathcal{A}$ naturally restricts the number of samples it can "see", resolving the apparent mystery.

An important follow-up work by Gupte et al. [39] reduced the well-known LWE problem to the Gaussian pancakes problem. Assuming sub-exponential hardness of LWE [52], a standard assumption underlying post-quantum cryptosystems expected to be standardized by NIST, the Gaussian pancakes problem is hard for any $\gamma \geq (\log d)^{1+\varepsilon}$, where $\varepsilon > 0$ is any constant, and $\sigma \geq 1/\mathrm{poly}(d)$ [39, Section 1.2]. Taken together, these findings strongly support the hardness of Gaussian pancakes for the specified regimes of $(\gamma, \sigma)$. Note, however, that the condition $\sigma \geq 1/\mathrm{poly}(d)$ is *necessary* for hardness as there exist polynomial-time algorithms, based on lattice basis reduction, for exponentially small $\sigma$ [83, 25].

## 3 Hardness of Score Estimation

Our main result is Theorem 3.1, which presents a reduction from the Gaussian pancakes problem to $L^2$-accurate score estimation. Since the Gaussian pancakes problem exhibits both cryptographic and SQ hardness in the parameter regime $\gamma \geq 2\sqrt{d}$ and $\sigma \geq 1/\mathrm{poly}(d)$ [13, 39], these notions of hardness extend to the task of estimating scores of Gaussian pancakes. Further details on the hardness of the Gaussian pancakes problem can be found in Section 2.3.

**Theorem 3.1** (Main result). *Let $\gamma(d) > 1, \sigma(d) > 0$ be any pair of sequences such that $\sigma \geq 1/\mathrm{poly}(d)$ and the corresponding (sequence of) $(\gamma, \sigma)$-Gaussian pancakes distributions $(P_{\boldsymbol{u}})_{\boldsymbol{u}\in\mathbb{S}^{d-1}}$*

*satisfies* $\mathrm{TV}(P_{\boldsymbol{u}}, Q_d) > 1/2$ *for any* $d \in \mathbb{N}$. *Then, for any* $\delta \in (0, 1)$, *there exists a* $\mathrm{poly}(d) \cdot \log(1/\delta)$-*time algorithm with access to a score estimation oracle of* $L^2$-*error* $O(1/\sqrt{\log d})$ *that solves the* $(\gamma, \sigma)$-*Gaussian pancakes problem with probability at least* $1 - \delta$.

The requirement $\mathrm{TV}(P_{\boldsymbol{u}}, Q_d) > 1/2$ is mild and entirely captures interesting parameter regimes of $(\gamma, \sigma)$ for which cryptographic and SQ hardness of Gaussian pancakes are known. We provide a sufficient condition in Lemma B.9, which shows that $\gamma\sigma < C$ for some constant $C > 0$ ensures separation in TV distance.

Theorem 3.1 implies that even a score estimation oracle running in time $\mathrm{poly}(d, 2^{1/\varepsilon^2})$, where $\varepsilon > 0$ is the $L^2$ estimation error bound, implies a $\mathrm{poly}(d)$ time algorithm for the Gaussian pancakes problem. This means that estimating the score functions of Gaussian pancakes to $L^2$-accuracy $\varepsilon$ even in $\mathrm{poly}(d, 2^{1/\varepsilon^2})$ time is impossible under standard cryptographic assumptions.

### 3.1 Proof outline of Theorem 3.1

Here, we sketch the proof of Theorem 3.1 and defer full details to Section B.1. We first recall the sampling-to-score estimation reduction of Chen et al. [17] and its required assumptions to illustrate the main idea behind our reduction. The precise formulation of our idea is given in Lemma 3.3.

**Assumptions on data distribution.** The reduction of Chen et al. [17] requires the following assumptions on the data distribution $D$ over $\mathbb{R}^d$.

**A1** (Lipschitz score). For all $t \geq 0$, the score $\nabla \log D_t$ is $L$-Lipschitz.

**A2** (Finite second moment). $D$ has finite second moment, i.e., $\mathfrak{m}_2^2 := \mathbb{E}_{\boldsymbol{x} \sim D}[\|\boldsymbol{x}\|^2] < \infty$.

**A3** (Score estimation error). For step size $h := T/N$ and all $k = 1, \ldots, N$,
$$\mathbb{E}_{D_{kh}}[\|s_{kh} - \nabla \log D_{kh}\|^2] \leq \varepsilon_{\mathrm{score}}^2 .$$

**Theorem 3.2** ([17, Theorem 2]). *Suppose assumptions* **A1**-**A3** *hold. Let* $Q_d$ *be the standard Gaussian on* $\mathbb{R}^d$ *and let* $F_T$ *be the output of the DDPM algorithm (Section 2.1) at time* $T$ *with step size* $h := T/N$ *such that* $h \lesssim 1/L$, *where* $L \geq 1$. *Then, it holds that*

$$\mathrm{TV}(F_T, D) \lesssim \underbrace{\sqrt{\mathrm{KL}(D \,\|\, Q_d)} \cdot \exp(-T)}_{\text{convergence of forward process}} + \underbrace{(L\sqrt{dh} + L\mathfrak{m}_2 h)\sqrt{T}}_{\text{discretization error}} + \underbrace{\varepsilon_{\mathrm{score}}\sqrt{T}}_{\text{score estimation error}} . \quad (6)$$

*In particular, if* $\mathfrak{m}_2 \leq d$, *then* $T \asymp \max(\log(\mathrm{KL}(D \,\|\, Q_d)/\varepsilon), 1)$ *and* $h \asymp \varepsilon^2/(L^2 d)$ *gives*

$$\mathrm{TV}(F_T, D) \lesssim \varepsilon + \varepsilon_{\mathrm{score}} \cdot \max(\sqrt{\log(\mathrm{KL}(D \,\|\, Q_d)/\varepsilon)}, 1) , \qquad \text{for } N = \tilde{\Theta}\Big(\frac{L^2 d}{\varepsilon^2}\Big) .$$

Theorem 3.2 shows that if the unknown data distribution $D$ has Lipschitz scores and satisfies $\mathfrak{m}_2 \leq d$, then its $\varepsilon$-accurate score estimates along the discretized forward process $(s_{kh})_{k \in [N]}$ can be used to compute a *certificate* of Gaussianity defined as follows.

$$\Delta := \max_{k \in [N]} \mathbb{E}_{Q_d}\|s_{kh}(\boldsymbol{x}) + 2\pi\boldsymbol{x}\|^2 . \quad (7)$$

Using $\Delta$ as a test statistic, we decide $D \in (P_{\boldsymbol{u}})_{\boldsymbol{u} \in \mathbb{S}^{d-1}}$ if $\Delta \geq \tau$ for some carefully chosen threshold $\tau > 0$ and $D = Q_d$ otherwise. This test statistic is motivated by the observation that the discretized reverse process depends on the data distribution $D$ *solely* through its score estimates (see Eq.(3)). For *any* sequence of score estimates $(s_t)_{t \in [0,T]}$ the reverse process outputs $F_T$ that is TV-close to $D$ provided its $L^2$ error along the forward process $(D_t)_{t \in [0,T]}$ is small.

We claim that if $\Delta \leq \eta^2$ *and* the score estimates $(s_{kh})$ are $\varepsilon$-accurate for $(D_{kh})$, then the output of the reverse process $F_T$ is roughly $(\varepsilon + \eta)$-close in TV distance to the standard Gaussian. This is because the standard Gaussian is invariant throughout the OU process, so $\Delta$ is, in fact, the $L^2$ score estimation error bound for the case where the data distribution $D$ is equal to $Q_d$. In other words, $\Delta \leq \eta^2$ means that for all $k \in [N]$, the score estimates $(s_{kh})$ are $\eta$-close to $-2\pi\boldsymbol{x}$ which is the score function of $Q_d$. Thus, $\Delta$ is small only if $D$ is close in TV distance to $Q_d$, which shows that $\Delta$ distinguishes between $D = Q_d$ and $D \in (P_{\boldsymbol{u}})_{\boldsymbol{u} \in \mathbb{S}^{d-1}}$ provided $\mathrm{TV}(P_{\boldsymbol{u}}, Q_d) > 1/2$. The following lemma formalizes this idea. Theorem 3.1 follows from Lemma 3.3 and Lipschitzness of the score functions (Lemma B.5).

**Lemma 3.3** (Gaussianity testing with scores). *For any $\varepsilon \in (0, 1)$ and $K \geq 2$, let $D$ be any distribution on $\mathbb{R}^d$ such that $\mathfrak{m}_2 \leq d$ and $\mathrm{KL}(D \parallel Q_d) \leq K$ with $L$-Lipschitz score $\nabla \log D_t$ for any $t \geq 0$, and let $\tilde{\varepsilon} \asymp \varepsilon / \sqrt{\log(K/\varepsilon)}$ be the $L^2$ score estimation error bound with discretization parameters $T \asymp \log(K/\varepsilon), h \asymp \varepsilon^2 / (L^2 d)$, and $N := T/h$ so that $\mathrm{TV}(F_T \parallel D) \leq \varepsilon$ (via Theorem 3.2). If $(s_{kh})_{k \in [N]}$ are $\tilde{\varepsilon}$-accurate score estimates for the forward process $(D_{kh})_{k \in [N]}$ and $\Delta = \max_{k \in [N]} \mathbb{E}_{Q_d} \|s_{kh}(\boldsymbol{x}) + 2\pi \boldsymbol{x}\|^2$, then*

$$\mathrm{TV}(D, Q_d) \lesssim \varepsilon + \sqrt{\Delta \log(K/\varepsilon)} \ .$$

*In particular, if $\mathrm{TV}(D, Q_d) > 1/2$ then there exist constants $C_1, C_2 > 0$ such that for any score estimates $(s_{kh})_{k \in [N]}$ of the forward process satisfying $\max_{k \in [N]} \mathbb{E}_{D_{kh}} \|s_{kh}(\boldsymbol{x}) - \nabla \log D_{kh}(\boldsymbol{x})\|^2 \leq C_1 / \log K$, it holds*

$$\Delta \geq C_2 / \log K \ .$$

*Proof.* By Theorem 3.2 and our choice of $L^2$ score estimation error bound $\tilde{\varepsilon}$, we have $\mathrm{TV}(F_T, D) \leq \varepsilon$. In addition, the score estimates $(s_t)$ also satisfy a $\sqrt{\Delta}$-error bound with respect to the forward process of $Q_d$, which is invariant with respect to time $t$. Thus, Theorem 3.2 applied with $D = Q_d$ as the data distribution, discretization parameters $T, h$, and score estimates $(s_{kh})$ gives $\mathrm{TV}(F_T, Q_d) \lesssim \sqrt{\Delta \log(K/\varepsilon)}$. By the triangle inequality, we have

$$\mathrm{TV}(D, Q) \leq \mathrm{TV}(F_T, D) + \mathrm{TV}(F_T, Q) \lesssim \varepsilon + \sqrt{\Delta \log(K/\varepsilon)} \ .$$

The second part of the theorem follows from using the assumptions $\mathrm{TV}(D, Q) > 1/2, K \geq 2$, and fixing $\varepsilon > 0$ to a sufficiently small constant, which gives us

$$1 \lesssim \sqrt{\Delta \log K} \ .$$

$\square$

# 4 Sample Complexity of Gaussian Pancakes

To establish that there is indeed a *gap* between statistical and computational feasibility, we demonstrate a polynomial upper bound on the sample complexity of $L^2$-accurate score estimation for Gaussian pancakes. In particular, we show that a sufficiently good estimate $\hat{\boldsymbol{u}}$ of the hidden direction $\boldsymbol{u}$ is enough (Lemma 4.1). The polynomial sample complexity of score estimation then follows from Theorem 4.2, which shows that if $\gamma(d)$ and $\sigma(d)$ satisfy $\gamma\sigma = O(1)$, then $1 - \langle \hat{\boldsymbol{u}}, \boldsymbol{u} \rangle^2 \leq \eta^2$ is *statistically* achievable with $\mathrm{poly}(d, \gamma, 1/\eta)$ samples, albeit through brute-force search over $\mathbb{S}^{d-1}$.

Our estimator $\hat{\boldsymbol{u}}$ is based on projection pursuit [35, 43]. We design a functional $E : \mathbb{S}^{d-1} \to \mathbb{R}$ of the form $E(\boldsymbol{v}) := \mathbb{E}_{\boldsymbol{x} \sim P_{\boldsymbol{u}}} g(\langle \boldsymbol{x}, \boldsymbol{v} \rangle)$ with $g : \mathbb{R} \to \mathbb{R}$ carefully chosen to ensure that $E(\boldsymbol{v}_1) \geq E(\boldsymbol{v}_2)$ if and only if $|\langle \boldsymbol{v}_1, \boldsymbol{u} \rangle| \geq |\langle \boldsymbol{v}_2, \boldsymbol{u} \rangle|$. Given such a "monotonic" functional (its empirical counterpart $\hat{E}$, to be precise), our estimator for the secret direction $\boldsymbol{u}$ is essentially

$$\hat{\boldsymbol{u}} = \arg\max_{\boldsymbol{v} \in \mathbb{S}^{d-1}} \hat{E}(\boldsymbol{v}) \ .$$

We remark that parameter regime $\gamma\sigma = O(1)$ in Theorem 4.2 encompasses the cryptographically hard regime of Gaussian pancakes. It is also worth noting that if $\min(\gamma, \gamma\sigma) = \omega(\sqrt{\log d})$, then Gaussian pancakes are *statistically* indistinguishable from $Q_d$ by Lemma B.10.

We defer the full proofs of Lemma 4.1 and Theorem 4.2 to Section C.

**Lemma 4.1** (Score-to-parameter estimation reduction). *For any $\gamma > 1, \sigma > 0$, let $P_{\boldsymbol{u}}$ be the $(\gamma, \sigma)$-Gaussian pancakes distribution with secret direction $\boldsymbol{u} \in \mathbb{S}^{d-1}$. Given any $\eta \in (0, 1)$ and $\hat{\boldsymbol{u}} \in \mathbb{S}^{d-1}$ such that $1 - \langle \hat{\boldsymbol{u}}, \boldsymbol{u} \rangle^2 \leq \eta^2$, the score estimate $\hat{s}(\boldsymbol{x}) = -2\pi \boldsymbol{x} + \nabla \log T_\gamma^\sigma(\langle \boldsymbol{x}, \hat{\boldsymbol{u}} \rangle)$ satisfies*

$$\mathbb{E}_{\boldsymbol{x} \sim P_{\boldsymbol{u}}} \|\hat{s}(\boldsymbol{x}) - s(\boldsymbol{x})\|^2 \lesssim \max(1, 1/\sigma^8) \cdot \eta^2 d \ ,$$

*where $s(\boldsymbol{x}) = -2\pi \boldsymbol{x} + \nabla \log T_\gamma^\sigma(\langle \boldsymbol{x}, \boldsymbol{u} \rangle)$ is the true score function of $P_{\boldsymbol{u}}$.*

**Theorem 4.2** (Sample complexity of parameter estimation). *For any constant $C > 0$, given $\gamma(d) > 1, \sigma(d) > 0$ such that $\gamma\sigma < C$, estimation error parameter $\eta > 0$, and $\delta \in (0, 1)$, there exists a brute-force search estimator $\hat{\boldsymbol{u}} : \mathbb{R}^{d \times n} \to \mathbb{S}^{d-1}$ that uses $n = \mathrm{poly}(d, \gamma, 1/\eta, 1/\delta)$ samples and achieves $\|\hat{\boldsymbol{u}}(\boldsymbol{x}_1, \ldots, \boldsymbol{x}_n) - \boldsymbol{u}\|^2 \leq \eta^2$ with probability at least $1 - \delta$ over i.i.d. samples $\boldsymbol{x}_1, \ldots, \boldsymbol{x}_n \sim P_{\boldsymbol{u}}$, where $P_{\boldsymbol{u}}$ is the $(\gamma, \sigma)$-Gaussian pancakes distribution with secret direction $\boldsymbol{u} \in \mathbb{S}^{d-1}$.*

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

# Appendices

## A  Additional Related Work

**Score estimation.**    Several works have addressed the statistical question of sample complexity for various data distributions and function classes used to estimate the score [16, 55, 80]. Wibisono et al. [80] study score estimation for the class of subgaussian distributions with Lipschitz scores. By establishing a minimax lower bound exhibiting a curse of dimensionality, they show that the exponential dependence on dimension is fundamental to this nonparametric distribution class, underscoring the need for stronger assumptions on the data distribution for polynomial sample complexity. Chen et al. [16] study neural network-based score estimation and derive finite-sample bounds for data distributions that lie on a low-dimensional linear subspace, which circumvents the curse of dimensionality. Mei and Wu [55] study neural network-based score estimation for graphical models which are intrinsically high dimensional. Assuming the efficiency of variational inference approximation to the data-generating graphical model, they show that the score function can be efficiently approximated by residual networks [40] and learned with polynomially many samples. Efficient score estimation both in sample and computational complexity has been achieved by Shah et al. [68] for mixtures of two spherical Gaussians using a shallow neural network architecture that matches the closed-form expression of the score function.

**Statistical-to-computational gaps.**    Statistical-to-computational gaps have a rich history in statistics, machine learning, and computer science [7, 85, 54, 69, 24]. Indeed, many high-dimensional inference problems exhibit gaps between what is achievable statistically (with infinite computational resources) and what is achievable under limited computation. Notable examples include sparse PCA [7], sparse linear regression [85, 79], and learning one-hidden-layer neural networks over Gaussian inputs [37, 30].

Since there are no known reductions from NP-hard problems, the gold standard for computational hardness, to any average-case problem arising in statistical inference, alternative techniques have been developed to provide rigorous evidence of hardness. These include proving lower bounds for restricted classes of algorithms like sum of squares (SoS) [49, 59, 5] and statistical query (SQ) algorithms [46, 34], which capture spectral, moment, and tensor methods, and reducing from "canonical" problems believed to be hard *on average*, such as the planted clique [11] or random $k$-SAT [21] problem. We refer the reader to surveys and monographs [84, 4, 81, 36, 48, 47, 29] for a thorough overview of recent literature and diverse perspectives ranging from computer science and information theory to statistical physics.

## B  Proofs for Section 3

### B.1  Proof of Theorem 3.1

**Theorem B.1** (Theorem 3.1 restated). *Let $\gamma(d) > 1, \sigma(d) > 0$ be any pair of sequences such that $\sigma \geq 1/\mathrm{poly}(d)$ and the corresponding (sequence of) $(\gamma, \sigma)$-Gaussian pancakes distributions $(P_{\boldsymbol{u}})_{\boldsymbol{u} \in \mathbb{S}^{d-1}}$ satisfies $\mathrm{TV}(P_{\boldsymbol{u}}, Q_d) > 1/2$ for any $d \in \mathbb{N}$. Then, for any $\delta \in (0, 1)$, there exists a $\mathrm{poly}(d) \cdot \log(1/\delta)$-time algorithm with access to a score estimation oracle of $L^2$-error $O(1/\sqrt{\log d})$ that solves the $(\gamma, \sigma)$-Gaussian pancakes problem with probability at least $1 - \delta$.*

*Proof.* Let $D \in (P_{\boldsymbol{u}}) \cup Q$ be the given data distribution. We first verify that assumptions A1-A3 hold, allowing us to apply the score-based Gaussianity test from Lemma 3.3. For any $\sigma(d), \gamma(d) > 0$ such that $\sigma \geq 1/\mathrm{poly}(d)$, the $(\gamma, \sigma)$-Gaussian pancakes satisfy the Lipschitz score condition $L \leq \mathrm{poly}(d)$ (Assumption A1) by Lemma B.5 and the second moment bound $\mathfrak{m}_2 \leq d$ (Assumption A2) by Lemma B.2. In addition, $\mathrm{KL}(P_{\boldsymbol{u}} \| Q) \leq \mathrm{poly}(d)$ (Lemma B.11), so Lemma 3.3 applies to $D$ with $K = \mathrm{poly}(d)$. Moreover, since $\mathrm{TV}(P_{\boldsymbol{u}}, Q) > 1/2$, Lemma 3.3 implies that if $D \in (P_{\boldsymbol{u}})_{\boldsymbol{u} \in \mathbb{S}^{d-1}}$, there exist universal constants $C_1, C_2 > 0$ such that if the score estimates $(s_{kh})_{k \in [N]}$ satisfy $\mathbb{E}_{D_{kh}} \| s_{kh}(\boldsymbol{x}) - \nabla \log D_{kh}(\boldsymbol{x}) \|^2 \leq C_1 / \log d$, then $\Delta \geq C_2 / \log d$.

Let $\tau = C_2 / \log d$ and $\eta^2 = \min(\tau/4, C_1 / \log d)$. Then, we have that with $\eta$-accurate score estimates $(s_{kh})$, we have $\Delta \leq \eta^2 \leq \tau/4$ if $D = Q$ and $\Delta \geq \tau$ otherwise. Note that $\eta \asymp 1/\sqrt{\log d}$ and $N = T/h \asymp L^2 d \log K \leq \mathrm{poly}(d)$. Our proposed distinguisher $\mathcal{A}$ uses a finite-sample estimate of $\Delta$ as a test statistic (Eq.(7)) using $\eta$-accurate score estimates $(s_{kh})_{k \in [N]}$ and $N\ell$ i.i.d. standard Gaussian samples $(\boldsymbol{z}_i^{(k)})_{(k,i) \in [N] \times [\ell]}$ as follows. Later, it will be shown that setting the batch size $\ell$

to $\ell = \text{poly}(d)$ is sufficient for our distinguisher $\mathcal{A}$.

$$\hat{\Delta} = \max_{k \in [N]} \hat{\Delta}^{(k)} , \quad \text{where } \hat{\Delta}^{(k)} = \frac{1}{\ell} \sum_{i=1}^{\ell} \left\| s_{kh}(z_i^{(k)}) + 2\pi z_i^{(k)} \right\|^2 .$$

The distinguisher $\mathcal{A}$ decides $D = Q$ if $\hat{\Delta} \leq \tau/2$ and $D \in (P_{\boldsymbol{u}})_{\boldsymbol{u} \in \mathbb{S}^{d-1}}$ otherwise. This procedure runs in time $O(N\ell)$ and makes $N = \text{poly}(d)$ queries to the score estimation oracle of $L^2$-accuracy $O(1/\sqrt{\log d})$.

One issue in estimating $\Delta$ is that the score estimates $(s_{kh})$ only satisfy the mean guarantee with respect to the forward process $(D_{kh})$, i.e., $\mathbb{E}_{D_{kh}} \|s_{kh}(\boldsymbol{x}) - \nabla \log D_{kh}(\boldsymbol{x})\|^2 \leq \eta^2$. These guarantees do not necessarily provide control over the concentration of random variables $\|s_t(\boldsymbol{z}) + 2\pi \boldsymbol{z}\|^2$ induced by $\boldsymbol{z} \sim Q_d$. Moreover, if $D = P_{\boldsymbol{u}}$, then $s_t(\boldsymbol{z})$ may behave erratically for $\boldsymbol{z} \sim Q_d$, taking on large norms in low density areas between the pancakes, which may deter the estimation of $\Delta$. We handle this by truncating the score estimates.

Let $M > 0$ be some large number to be determined later. Define the truncated score $\bar{s}$ by

$$\bar{s}_t(\boldsymbol{x}) = \begin{cases} s_t(\boldsymbol{x}) & \text{if } \|s_t(\boldsymbol{x})\| \leq M , \\ \boldsymbol{0} & \text{otherwise} . \end{cases}$$

We claim that using the truncated score estimates $(\bar{s}_t)$ in place of $(s_t)$ introduces negligible (in data dimension $d$) additional $L^2$ score estimation error with respect to the forward process $(D_t)$ compared to the original score estimates $(s_t)$. Hence, Lemma 3.3 applies with the *uniformly bounded* vector fields $(\tilde{s}_{kh})_{k \in [N]}$ as the $L^2$-accurate score estimates for $(D_{kh})_{k \in [N]}$. For any (discretized) time $0 \leq t \leq T$ and distribution $D_t$ from the forward process,

$$\mathbb{E}_{D_t} \|\bar{s}_t(\boldsymbol{x}) - \nabla \log D_t(\boldsymbol{x})\|^2 = \mathbb{E}_{D_t} \big[ \|s_t(\boldsymbol{x}) - \nabla \log D_t(\boldsymbol{x})\|^2 \cdot \mathbb{1}[\|s_t(\boldsymbol{x})\| \leq M] \big]$$
$$+ \mathbb{E}_{D_t} \big[ \|\nabla \log D_t(\boldsymbol{x})\|^2 \cdot \mathbb{1}[\|s_t(\boldsymbol{x})\| > M] \big] .$$

The second term on the RHS can be upper bounded by

$$\mathbb{E}_{D_t} \big[ \|\nabla \log D_t(\boldsymbol{x})\|^2 \cdot \mathbb{1}[\|s_t(\boldsymbol{x})\| > M] \big]$$
$$= \mathbb{E}_{D_t} \big[ \|\nabla \log D_t(\boldsymbol{x})\|^2 \cdot \mathbb{1}[(\|s_t(\boldsymbol{x})\| > M) \wedge (\|\nabla \log D_t(\boldsymbol{x})\| > M/2)] \big]$$
$$+ \mathbb{E}_{D_t} \big[ \|\nabla \log D_t(\boldsymbol{x})\|^2 \cdot \mathbb{1}[(\|s_t(\boldsymbol{x})\| > M) \wedge (\|\nabla \log D_t(\boldsymbol{x})\| \leq M/2)] \big]$$
$$\leq \mathbb{E}_{D_t} \big[ \|\nabla \log D_t(\boldsymbol{x})\|^2 \cdot \mathbb{1}[\|\nabla \log D_t(\boldsymbol{x})\| > M/2] \big]$$
$$+ \mathbb{E}_{D_t} \big[ \|s_t(\boldsymbol{x}) - \nabla \log D_t(\boldsymbol{x})\|^2 \cdot \mathbb{1}[\|s_t(\boldsymbol{x})\| > M] \big] , \tag{8}$$

where in Eq.(8) we used the fact that if $\|s_t(\boldsymbol{x})\| > M$ and $\|\nabla \log D_t(\boldsymbol{x})\| \leq M/2$, then $\|\nabla \log D_t(\boldsymbol{x})\| \leq M/2 \leq \|s_t(\boldsymbol{x}) - \nabla \log D_t(\boldsymbol{x})\|$.

Putting things together, we have

$$\mathbb{E}_{D_t} \|\bar{s}_t(\boldsymbol{x}) - \nabla \log D_t(\boldsymbol{x})\|^2 \leq \mathbb{E}_{D_t} \|s_t(\boldsymbol{x}) - \nabla \log D_t(\boldsymbol{x})\|^2$$
$$+ \mathbb{E}_{D_t} \big[ \|\nabla \log D_t(\boldsymbol{x})\|^2 \cdot \mathbb{1}[\|\nabla \log D_t(\boldsymbol{x})\| > M/2] \big] .$$

It remains to bound $\mathbb{E}[\|\nabla \log D_t(\boldsymbol{x})\|^2 \cdot \mathbb{1}[\|\nabla \log D_t(\boldsymbol{x})\| > M/2]]$ which depends only on the distribution $D_t$. We choose $M \asymp (\sqrt{d} + 1/\sigma^2)$. If $D = Q_d$, then $D_t = Q_d$ for any $t > 0$, and by Cauchy-Schwarz and norm concentration (see e.g., [77, Theorem 3.1.1]), there exists a constant $C > 0$ such that

$$\mathbb{E}_{Q_d} \big[ \|\boldsymbol{x}\|^2 \cdot \mathbb{1}[\|\boldsymbol{x}\| > M/2] \big] \leq \mathbb{E}_{Q_d} \|\boldsymbol{x}\|^4 \cdot \Pr_{Q_d}[\|\boldsymbol{x}\| > M/2] \leq \exp(-CM^2) .$$

On the other hand, if $D = P_{\boldsymbol{u}}$, then by Lemma B.4, Eq.(13) and the triangle inequality

$$\|\nabla \log P_{\boldsymbol{u}}(\boldsymbol{x})\| = \left\| \boldsymbol{x} + \frac{(T_\gamma^\sigma)'(\langle \boldsymbol{x}, \boldsymbol{u} \rangle)}{T_\gamma^\sigma(\langle \boldsymbol{x}, \boldsymbol{u} \rangle)} \boldsymbol{u} \right\| \leq \|\boldsymbol{x}\| + 8\pi(1 + 1/\sigma^2) .$$

Using a similar concentration argument, we have

$$\mathbb{E}_{P_{\boldsymbol{u}}}\left[\|\nabla \log P_{\boldsymbol{u}}(\boldsymbol{x})\|^2 \cdot \mathbb{1}[\|\nabla \log P_{\boldsymbol{u}}(\boldsymbol{x})\| > M/2]\right]$$
$$\lesssim \mathbb{E}_{P_{\boldsymbol{u}}}\left[(\|\boldsymbol{x}\|^2 + 1/\sigma^4 + 1) \cdot \mathbb{1}[\|\boldsymbol{x}\| > M/2 - 8\pi(1 + 1/\sigma^2)]\right]$$
$$\lesssim \exp(-O(M^2)) . \tag{9}$$

The OU process applied to $P_{\boldsymbol{u}}$ only increases the $\sigma$ parameter, so the upper bound in Eq.(9) holds uniformly over the forward process $(D_{kh})_{k \in [N]}$ if $D = P_{\boldsymbol{u}}$. Thus, choosing $M \asymp (\sqrt{d} + 1/\sigma^2) = \mathrm{poly}(d)$ as the truncation threshold suffices to ensure that for all discretized time steps $t = kh$, the $L^2$ score estimation error with respect to the forward process $(D_t)$ introduced by truncating the score estimate $s_t$ to $\bar{s}_t$ is negligible in $d$. Therefore, we apply Lemma 3.3 with $(\tilde{s}_{kh})_{k \in [N]}$ as the score estimates for the forward process $(D_{kh})_{k \in [N]}$.

Since $\|\bar{s}_t(\boldsymbol{z}) + 2\pi\boldsymbol{z}\|^2$, where $\boldsymbol{z} \sim Q_d$, is a random variable with subexponential norm $O(M^2)$, we can apply Bernstein's inequality [77, Corollary 2.8.3] to the i.i.d. sum $\hat{\Delta}^{(k)}$. Thus, for any $k \in [N]$, $\ell \asymp (M^4/\varepsilon^2) \cdot \log(N/\delta)$ Gaussian samples suffice to guarantee accurate estimation of the population mean $\Delta^{(k)}$ with additive error less than $\varepsilon$ with probability at least $1 - \delta/N$. By a union bound, with probability at least $1 - \delta$, this holds for all $k \in [N]$. Setting $\varepsilon = \tau/8 = \Theta(1/\log d)$ and recalling that $N = \mathrm{poly}(d)$, we have a distinguisher $\mathcal{A}$ for the Gaussian pancakes problem that makes $N = \mathrm{poly}(d)$ queries to the score estimation oracle, runs in time $N\ell = \mathrm{poly}(d) \cdot \log(1/\delta)$, and is correct with probability at least $1 - \delta$. $\qquad\square$

**Lemma B.2** (Second moment of Gaussian pancakes). *For any $\gamma, \sigma > 0$, and $\boldsymbol{u} \in \mathbb{S}^{d-1}$, the $(\gamma, \sigma)$-Gaussian pancakes $P_{\boldsymbol{u}}$ satisfies*

$$\mathbb{E}_{\boldsymbol{x} \sim P_{\boldsymbol{u}}}[\|\boldsymbol{x}\|^2] \leq \frac{d}{2\pi} .$$

*Proof.* Without loss of generality, we assume $\boldsymbol{u} = \boldsymbol{e}_1$. Then,

$$\mathbb{E}_{\boldsymbol{x} \sim P_{\boldsymbol{u}}}\|\boldsymbol{x}\|^2 = \mathbb{E}_{x \sim A_\gamma^\sigma}[x^2] + (d-1)\mathbb{E}_{z \sim Q}z^2 = \mathbb{E}_{x \sim A_\gamma^\sigma}[x^2] + \frac{d-1}{2\pi} .$$

In addition,

$$\mathbb{E}_{x \sim A_\gamma^\sigma}[x^2] = \mathbb{E}_{x \sim A_\gamma}\mathbb{E}_{z \sim \mathcal{N}(0, 1/(2\pi))}\left(x/\sqrt{1 + \sigma^2} + \sigma z/\sqrt{1 + \sigma^2}\right)^2$$
$$= \frac{1}{1 + \sigma^2} \cdot \mathbb{E}_{x \sim A_\gamma}[x^2] + \frac{\sigma^2}{1 + \sigma^2} \cdot \frac{1}{2\pi} .$$

Thus, it suffices to establish an upper bound on $\mathbb{E}_{x \sim A_\gamma} x^2$, i.e., the second moment of the discrete Gaussian on $(1/\gamma)\mathbb{Z}$. Using the Poisson summation formula (Lemma 2.1) and the fact that the Fourier transform of $x^2 \rho(x)$ is $(1/(2\pi) - y^2)\rho(y)$,

$$\mathbb{E}_{x \sim A_\gamma}[x^2] = \frac{1}{\rho((1/\gamma)\mathbb{Z})} \sum_{x \in (1/\gamma)\mathbb{Z}} x^2 \rho(x)$$
$$= \frac{\gamma}{\rho((1/\gamma)\mathbb{Z})} \sum_{y \in \gamma\mathbb{Z}} \left(\frac{1}{2\pi} - y^2\right)\rho(y)$$
$$= \frac{1}{2\pi} \cdot \frac{\gamma}{\rho((1/\gamma)\mathbb{Z})} \cdot \rho(\gamma\mathbb{Z}) - \frac{\gamma}{\rho((1/\gamma)\mathbb{Z})} \sum_{y \in \gamma\mathbb{Z}} y^2 \rho(y) \tag{10}$$
$$< \frac{1}{2\pi}$$

where in Eq.(10), we used the fact that $\rho(\gamma\mathbb{Z}) = \rho((1/\gamma)\mathbb{Z})/\gamma$ and that the second term is positive.

Since $\mathbb{E}_{x \sim A_\gamma^\sigma}[x^2]$ is a convex combination of $\mathbb{E}_{x \sim A_\gamma}[x^2]$ and $1/(2\pi)$, the conclusion follows.

$\qquad\square$

## B.2 Score functions of Gaussian pancakes

In this section, we show that score functions of Gaussian pancakes distributions are Lipschitz with respect to $\boldsymbol{x} \in \mathbb{R}^d$. The score function of $P_{\boldsymbol{u}}$, the $(\gamma, \sigma)$-Gaussian pancakes distribution with secret direction $\boldsymbol{u} \in \mathbb{S}^{d-1}$, admits the following analytical expression.

$$\nabla \log P_{\boldsymbol{u}}(\boldsymbol{x}) = -2\pi\boldsymbol{x} + \nabla \log T(\langle \boldsymbol{x}, \boldsymbol{u} \rangle) = -2\pi\boldsymbol{x} + \frac{(T_\gamma^\sigma)'(\langle \boldsymbol{x}, \boldsymbol{u} \rangle)}{T_\gamma^\sigma(\langle \boldsymbol{x}, \boldsymbol{u} \rangle)} \boldsymbol{u} \ . \tag{11}$$

We use Banaszczyk's theorems on the Gaussian mass of lattices [3] to upper bound the Lipschitz constant of $(T_\gamma^\sigma)'/T_\gamma^\sigma$ in terms of $\sigma$.

**Theorem B.3** ([74, Corollary 1.3.5]). *For any lattice $\mathcal{L} \subset \mathbb{R}^d$, parameter $s > 0$, shift $\boldsymbol{t} \in \mathbb{R}^d$, and radius $r > \sqrt{d/(2\pi)} \cdot s$,*

$$\rho_s((\mathcal{L} - \boldsymbol{t}) \setminus r\mathcal{B}_2^d) < \exp(-\pi x^2)\rho_s(\mathcal{L}) \ ,$$

*where $x := r/s - \sqrt{d/(2\pi)}$ and $\mathcal{B}_2^d$ denotes the Euclidean ball in $\mathbb{R}^d$.*

**Lemma B.4** ([74, Lemma 1.3.10]). *For any lattice $\mathcal{L} \subset \mathbb{R}^d$, parameter $s > 0$, and shift $\boldsymbol{t} \in \mathbb{R}^d$,*

$$\exp(-\pi \cdot \operatorname{dist}(\boldsymbol{t}, \mathcal{L})^2 / s^2) \cdot \rho_s(\mathcal{L}) \leq \rho_s(\mathcal{L} - \boldsymbol{t}) \leq \rho_s(\mathcal{L}) \ .$$

**Lemma B.5** (Lipschitzness of $\nabla \log P_{\boldsymbol{u}}$). *For any $\gamma > 1, \sigma > 0$, $s = \sigma/\sqrt{1+\sigma^2}$, and $\boldsymbol{u} \in \mathbb{S}^{d-1}$, the score function of the $(\gamma, \sigma)$-Gaussian pancakes distribution $P_{\boldsymbol{u}}$ satisfies the Lipschitz condition:*

$$\|\nabla \log P_{\boldsymbol{u}}(\boldsymbol{y}) - \nabla \log P_{\boldsymbol{u}}(\boldsymbol{x})\| \lesssim (1/s^4)\|\boldsymbol{y} - \boldsymbol{x}\| \quad \text{for any } \boldsymbol{x}, \boldsymbol{y} \in \mathbb{R}^d \ , \tag{12}$$

*and the likelihood ratio $T_\gamma^\sigma$ of the $\sigma$-smoothed discrete Gaussian $A_\gamma^\sigma$ relative to $\mathcal{N}(0, 1/(2\pi))$ satisfies the uniform bound:*

$$\left| \frac{(T_\gamma^\sigma)'(z)}{T_\gamma^\sigma(z)} \right| \leq \frac{8\pi}{s^2} \quad \text{for any } z \in \mathbb{R} \ . \tag{13}$$

*Proof.* It suffices to analyze the Lipschitz constant of the univariate function $(T_\gamma^\sigma)'/T_\gamma^\sigma : \mathbb{R} \to \mathbb{R}$, which we denote by $f = (T_\gamma^\sigma)'/T_\gamma^\sigma$ since for any $\boldsymbol{x} \neq \boldsymbol{y} \in \mathbb{R}^d$,

$$\begin{aligned}
\|\nabla \log P_{\boldsymbol{u}}(\boldsymbol{y}) - \nabla \log P_{\boldsymbol{u}}(\boldsymbol{x})\| &\leq 2\pi\|\boldsymbol{y} - \boldsymbol{x}\| + \big\|(f(\langle \boldsymbol{y}, \boldsymbol{u} \rangle) - f(\langle \boldsymbol{x}, \boldsymbol{u} \rangle))\boldsymbol{u}\big\| \\
&\leq 2\pi\|\boldsymbol{y} - \boldsymbol{x}\| + \frac{|f(\langle \boldsymbol{y}, \boldsymbol{u} \rangle) - f(\langle \boldsymbol{x}, \boldsymbol{u} \rangle)|}{|\langle \boldsymbol{y}, \boldsymbol{u} \rangle - \langle \boldsymbol{x}, \boldsymbol{u} \rangle|} \cdot |\langle \boldsymbol{y} - \boldsymbol{x}, \boldsymbol{u} \rangle| \\
&\leq \left( 2\pi + \sup_{a,b \in \mathbb{R}} \left| \frac{f(b) - f(a)}{b - a} \right| \right) \|\boldsymbol{y} - \boldsymbol{x}\| \ .
\end{aligned}$$

We bound the Lipschitz constant of the function $f(z)$ by demonstrating a uniform upper bound on the absolute value of its derivative $f'(z) := (d/dz)f(z)$. For notational convenience, we omit the parameters $\gamma, \sigma$ when denoting the likelihood ratio $T$. The derivative $f'$ is given by

$$f' = \left( \frac{T'}{T} \right)' = \frac{(T'')T - (T')^2}{T^2} = \frac{T''}{T} - \left( \frac{T'}{T} \right)^2 \ . \tag{14}$$

Hence, $|f'(z)| \leq |(T''/T)(z)| + |(T'/T)(z)|^2$ for any $z \in \mathbb{R}$. We prove uniform upper bounds on the two RHS terms, starting with $|T'/T|$. Using the definition of the likelihood ratio $T$ (Eq.(5)), we have

$$\begin{aligned}
\frac{T'(z)}{T(z)} &= \frac{2\pi}{\sigma^2} \cdot \frac{\sum_{k \in \mathbb{Z}} -(z - \sqrt{1+\sigma^2}k/\gamma)\rho_\sigma(z - \sqrt{1+\sigma^2}k/\gamma)}{\sum_{k \in \mathbb{Z}} \rho_\sigma(z - \sqrt{1+\sigma^2}k/\gamma)} \\
&= \frac{2\pi}{\sigma s} \cdot \underset{x \sim D_{\mathcal{L} - \tilde{z}, s}}{\mathbb{E}} [x] \ ,
\end{aligned} \tag{15}$$

where $\tilde{z} = z/\sqrt{1+\sigma^2}$, $s = \sigma/\sqrt{1+\sigma^2}$, $\mathcal{L} = (1/\gamma)\mathbb{Z}$, and $D_{\mathcal{L}-\tilde{z},s}$ is the discrete Gaussian distribution of width $s$ on the lattice coset $\mathcal{L} - \tilde{z}$.

We upper bound $\mathbb{E}[|x|]$ via a tail bound on $D_{\mathcal{L}-\tilde{z},s}$. Let $r > 0$ be any number and denote $t := r/s - \sqrt{1/(2\pi)}$. By Theorem B.3 and Lemma B.4,

$$
\begin{aligned}
\rho_s((\mathcal{L} - \tilde{z}) \setminus r\mathcal{B}_2) &< \exp(-\pi t^2)\rho_s(\mathcal{L}) \\
&< \exp(-\pi t^2)\exp(\pi \cdot \mathrm{dist}(\tilde{z}, \mathcal{L})^2/s^2)\rho_s(\mathcal{L} - \tilde{z}) \\
&= \exp(-\pi((r/s - \sqrt{1/(2\pi)})^2 - 1/(2s\gamma)^2))\rho_s(\mathcal{L} - z) ,
\end{aligned}
$$

where we used the fact that $\mathrm{dist}(\tilde{z}, \mathcal{L}) \leq 1/(2\gamma)$ for any $\tilde{z} \in \mathbb{R}$ in the last line.

Thus, for $r/(2s) \geq \sqrt{1/(2\pi)} + 1/(2s\gamma)$,

$$
\Pr_{x \sim D_{\mathcal{L}-\tilde{z},s}}[|x| \geq r] = \frac{\rho_s((\mathcal{L} - \tilde{z}) \setminus r\mathcal{B}_2)}{\rho_s(\mathcal{L} - \tilde{z})} \leq \exp(-\pi r^2/(2s)^2) . \tag{16}
$$

Denote $r_0 := s\sqrt{2/\pi} + 1/\gamma$. Note that $r_0 < \sqrt{2/\pi} + 1/\gamma$ since $s \in [0, 1)$. Then,

$$
\begin{aligned}
\mathbb{E}_{x \sim D_{\mathcal{L}-\tilde{z},s}}|x| = \int_r \Pr[|x| \geq r]dr &\leq r_0 + \int_{r>r_0} \Pr[|x| \geq r]dr \\
&\leq r_0 + \int_{r>r_0} \exp(-\pi r^2/(2s)^2)dr \\
&\leq r_0 + 2s \\
&\leq (2 + \sqrt{2/\pi})s + 1/\gamma . \tag{17}
\end{aligned}
$$

Therefore by Eq.(15) and (17), for any $z \in \mathbb{R}$

$$
\left|\frac{T'(z)}{T(z)}\right| \leq \frac{2\pi}{\sigma s}((2 + \sqrt{2/\pi})s + 1/\gamma) . \tag{18}
$$

Eq.(13) in the statement of Lemma B.5 follows immediately from the fact that $s < \min(1, \sigma)$ and $\gamma > 1$. Next, we demonstrate a uniform upper bound on $T''/T$. The analytical expression of $T''/T$ is given by

$$
\begin{aligned}
\frac{T''(z)}{T(z)} &= \left(\frac{2\pi}{\sigma^2}\right)^2 \cdot \left(\frac{\sum_{k \in \mathbb{Z}}(z - \sqrt{1+\sigma^2}k/\gamma)^2\rho_\sigma(z - \sqrt{1+\sigma^2}k/\gamma)}{\sum_{k \in \mathbb{Z}}\rho_\sigma(z - \sqrt{1+\sigma^2}k/\gamma)} - \frac{\sigma^2}{2\pi}\right) \\
&= \left(\frac{2\pi}{\sigma s}\right)^2 \left(\mathbb{E}_{x \sim D_{\mathcal{L}-\tilde{z},s}} x^2 - \frac{s^2}{2\pi}\right) ,
\end{aligned}
$$

where $\tilde{z} = z/\sqrt{1+\sigma^2}$, $s = \sigma/\sqrt{1+\sigma^2}$, and $\mathcal{L} = (1/\gamma)\mathbb{Z}$.

To uniformly bound $|T''/T|$, we upper bound the second moment of $D_{\mathcal{L}-\tilde{z},s}$. Again, let $r_0 = s\sqrt{2/\pi} + 1/\gamma$. Using the tail bound from Eq.(16) and the fact that $(a + b)^2 \leq 2a^2 + 2b^2$ for any $a, b \in \mathbb{R}$

$$
\begin{aligned}
\mathbb{E}_{x \sim D_{\mathcal{L}-z,\sigma}} x^2 = \int_0^\infty r\Pr[|x| \geq r]dr &\leq r_0^2/2 + \int_{r \geq r_0} r\Pr[|x| \geq r]dr \\
&\leq r_0^2/2 + \int_{r \geq r_0} r\exp(-\pi r^2/(2s^2))dr \\
&\leq r_0^2/2 + s^2/\pi \\
&\leq s^2 + 1/\gamma^2 . \tag{19}
\end{aligned}
$$

Applying Eq.(17) and Eq.(19) to the expression for $f' = T'/T$ (Eq.(14)) and using the fact that $\gamma > 1$ and $s = \sigma/\sqrt{1+\sigma^2} \leq \min(1,\sigma)$, we have

$$
\begin{aligned}
\|f'\|_\infty &\leq \|T''/T\|_\infty + \|(T'/T)^2\|_\infty \\
&\leq \left(\frac{2\pi}{\sigma s}\right)^2 \left((1+1/2\pi)s^2 + 1/\gamma^2 + ((2+\sqrt{2/\pi})s + 1/\gamma)^2\right) \\
&\leq \frac{100\pi^2}{s^4} \ .
\end{aligned}
$$

Therefore, $\nabla \log P_{\boldsymbol{u}}(\boldsymbol{x})$ is $O(1/s^4)$-Lipschitz since $2\pi + \|f'\|_\infty \lesssim 1/s^4$. $\qquad\square$

## B.3 Distance from the standard Gaussian

We now prove lower (Lemma B.9) and upper (Lemma B.10) bounds on the TV distance between Gaussian pancakes distributions $(P_{\boldsymbol{u}})$ and the standard Gaussian $Q$. We also show that the KL divergence is upper bounded by $\mathrm{poly}(d)$ for Gaussian pancakes with $\sigma \geq 1/\mathrm{poly}(d)$ (Lemma B.11).

The following fact reduces the $d$-dimensional problem of bounding $\mathrm{TV}(P_{\boldsymbol{u}}, Q_d)$ to a one-dimensional problem of bounding $\mathrm{TV}(A_\gamma^\sigma, Q)$, where $A_\gamma^\sigma$ is the $\sigma$-smoothed discrete Gaussian on $(1/\gamma)\mathbb{Z}$ and $Q = Q_1$. Without loss of generality, assume $\boldsymbol{u} = \boldsymbol{e}_1$. Then, by the $L^1$-characterization of the TV distance, we have

$$
\begin{aligned}
\mathrm{TV}(P_{\boldsymbol{u}}, Q_d) = \frac{1}{2}\int |P_{\boldsymbol{u}}(\boldsymbol{x}) - Q_d(\boldsymbol{x})|d\boldsymbol{x} &= \frac{1}{2}\int Q_d(\boldsymbol{x})|T_\gamma^\sigma(x_1) - 1|d\boldsymbol{x} \\
&= \frac{1}{2}\int |A_\gamma^\sigma(x_1) - Q(x_1)|dx_1 \\
&= \mathrm{TV}(A_\gamma^\sigma, Q) \ .
\end{aligned}
$$

Hence, it suffices to demonstrate bounds on $\mathrm{TV}(A_\gamma^\sigma, Q)$. The same applies to the KL divergence since $\mathrm{KL}(P_{\boldsymbol{u}} \parallel Q_d) = \mathrm{KL}(A_\gamma^\sigma \parallel Q)$. We first demonstrate a lower bound on the total variation distance. To this end, we introduce the periodic Gaussian *distribution* and its useful properties. The key lemma is Lemma B.9.

**Definition B.6** (Periodic Gaussian distribution). *For any one-dimensional lattice $\mathcal{L} \subset \mathbb{R}$, we define the periodic Gaussian* distribution $\Psi_{\mathcal{L},s} : \mathbb{R} \to \mathbb{R}_{\geq 0}$ *as follows.*

$$
\Psi_{\mathcal{L},s}(z) := \frac{1}{s}\sum_{x \in \mathcal{L}} \rho_s(x - z) = \rho_s(\mathcal{L} - z)/s \ .
$$

We can regard the function $\Psi_{\mathcal{L},s}$ as a distribution for the following reason: Let $\lambda_1(\mathcal{L})$ denote the spacing of $\mathcal{L}$. Then, $\Psi_{\mathcal{L},s}$ restricted to $[0, \lambda_1(\mathcal{L})]$ is a probability density since

$$
\int_0^{\lambda_1(\mathcal{L})} \Psi_{\mathcal{L},s}(z)dz = \frac{1}{s}\int_0^{\lambda_1(\mathcal{L})} \sum_{x \in \mathcal{L}} \rho_s(x - z) = \frac{1}{s}\sum_{x \in \mathcal{L}} \int_0^{\lambda_1(\mathcal{L})} \rho_s(x - z)dz = \frac{1}{s}\int_{-\infty}^{\infty} \rho_s(z)dz = 1 \ .
$$

**Lemma B.7** (Mill's inequality [77, Proposition 2.1.2]). *Let $z \sim \mathcal{N}(0,1)$. Then for all $t > 0$, we have*

$$
\mathbb{P}(|z| \geq t) = \sqrt{\frac{2}{\pi}}\int_t^{\infty} e^{-x^2/2}dx \leq \frac{1}{t}\cdot\sqrt{\frac{2}{\pi}}e^{-t^2/2} \ .
$$

**Lemma B.8** (Periodic Gaussian density bound [71, Claim I.6]). *For any $s > 0$ and any $z \in [0,1)$ the periodic Gaussian density $\Psi_{\mathbb{Z},s} : [0,1) \to \mathbb{R}_{\geq 0}$ satisfies*

$$
|\Psi_{\mathbb{Z},s}(z) - 1| \leq 2(1 + 1/(\pi s))e^{-\pi s^2} \ .
$$

*Proof.* By the Poisson summation formula (Lemma 2.1),

$$\Psi_{\mathbb{Z},s}(z) = \frac{1}{s}\sum_{x\in\mathbb{Z}}\rho_s(x-z)$$

$$= \sum_{y\in\mathbb{Z}} e^{-2\pi izy}\cdot\rho_{1/s}(y)$$

$$= 1 + \sum_{y\in\mathbb{Z}\setminus\{0\}} e^{-2\pi izy}\cdot\rho_{1/s}(y)\ .$$

Since $|e^{ia}| \le 1$ for any $a \in \mathbb{R}$, we have

$$|\Psi_{\mathbb{Z},s}(z) - 1| \le \sum_{y\in\mathbb{Z}\setminus\{0\}} |e^{-2\pi izy}|\cdot\rho_{1/s}(y) \le \sum_{y\in\mathbb{Z}\setminus\{0\}} \rho_{1/s}(y)$$

$$\le 2\left(e^{-\pi s^2} + \int_1^\infty e^{-\pi s^2 t^2}dt\right)$$

$$\le 2(1 + 1/(\pi s))e^{-\pi s^2}\ .$$

$\square$

**Lemma B.9** (TV lower bound). *There exists a constant $C > 0$ such that for any $\gamma > 1$ and $\sigma > 0$, if $\gamma\sigma < C$, then $\mathrm{TV}\big(A_\gamma^\sigma, \mathcal{N}(0, 1/(2\pi))\big) > 1/2$, where $A_\gamma^\sigma$ is the $\sigma$-smoothed discrete Gaussian on $(1/\gamma)\mathbb{Z}$.*

*Proof.* For ease of notation, we denote $\mathcal{N}(0, 1/(2\pi))$ by $Q$. Since $\mathrm{TV}(A_\gamma^\sigma, Q) = \sup_{S\in\mathcal{F}} |A_\gamma^\sigma(S) - Q(S)|$, where $\mathcal{F}$ is the Borel $\sigma$-algebra on $\mathbb{R}$, it suffices to find a measurable set $S \subset \mathbb{R}$ such that $A_\gamma^\sigma(S) - Q(S) > 1/2$.

Let $C = 1/(12\sqrt{2\log 2})$ be the constant in the statement of Lemma B.9. Let $\delta \in (0, 1/4)$ be the smallest number satisfying the condition $\gamma\sigma \le \delta/(3\sqrt{\log(1/\delta)})$. Such $\delta > 0$ always exists under the given assumptions since $\delta/\sqrt{\log(1/\delta)}$ is increasing in $\delta$ and $\delta = 1/4$ satisfies the condition (thanks to our choice of $C$). We claim that the set $S$ defined below witnesses the TV lower bound.

$$S := \left\{z \in \mathbb{R} \mid \mathrm{dist}(z, \mathcal{L}) \le \frac{\sigma}{\sqrt{1+\sigma^2}}\cdot\sqrt{\log(1/\delta)}\right\},$$

where $\mathcal{L} = (1/\gamma\sqrt{1+\sigma^2})\mathbb{Z}$.

We show a lower bound for $A_\gamma^\sigma(S)$ and an upper bound for $Q(S)$. Using the mixture form of the density of $A_\gamma^\sigma$ (Eq.(4)) and Mill's tail bounds for the univariate Gaussian (Lemma B.7), we have that for each Gaussian component in the mixture $A_\gamma^\sigma$, at least $1 - \delta$ fraction of its probability mass is contained in $S$. This is because the component means precisely form the one-dimensional lattice $\mathcal{L}$ and $S$ contains all significant neighborhoods of $\mathcal{L}$. Thus, $A_\gamma^\sigma(S) \ge 1 - \delta$.

Now we show an upper bound for $Q(S)$. Recall from Definition B.6 the density of the periodic Gaussian distribution $\Psi_{\mathbb{Z},s}$. Since $S$ is a periodic set, its mass $Q(S)$ is equal to $\Psi_{\mathbb{Z},s}(\tilde{S}\cap\mathbb{Z})$, where $s = \gamma\sqrt{1+\sigma^2}$ and

$$\tilde{S} = \left\{z \in \mathbb{R} \mid \mathrm{dist}(z, \mathbb{Z}) \le \gamma\sigma\sqrt{\log(1/\delta)}\right\}\ .$$

Since $s = \gamma\sqrt{1+\sigma^2} > 1$, by Lemma B.8 for any $z \in [0, 1)$

$$|\Psi_{\mathbb{Z},s}(z) - 1| \le 4e^{-\pi s^2} < 1/2\ .$$

Since $\gamma\sigma \le \delta/(3\sqrt{\log(1/\delta)})$, it follows that

$$Q(S) = \Psi_{\mathbb{Z},s}(\tilde{S}\cap[0,1]) \le \left(1 + 4e^{-\pi s^2}\right)\cdot 2\gamma\sigma\sqrt{\log(1/\delta)} < 3\gamma\sigma\sqrt{\log(1/\delta)} \le \delta\ .$$

Therefore,

$$\mathrm{TV}(A_\gamma^\sigma, Q) \ge A_\gamma^\sigma(S) - Q(S) > 1 - 2\delta > 1/2\ .$$

$\square$

We now establish upper bounds on $\mathrm{TV}(P_{\boldsymbol{u}}, Q_d)$ via upper bounds on $\mathrm{TV}(A_\gamma^\sigma, Q)$. Lemma B.10 provides a tighter upper bound when $\min(\gamma, \gamma\sigma) = \omega(\sqrt{\log d})$. In this regime, the sequence $(\mathrm{TV}(P_{\boldsymbol{u}}, Q_d))_{d \in \mathbb{N}}$ is *negligible* in $d$. It is worth noting that Lemma B.10 is not tight since $\mathrm{TV}(A_\gamma^\sigma, Q) \to 0$ as $\sigma \to \infty$, whereas the upper bound only converges to $e^{-\pi\gamma^2}$.

On the other hand, Lemma B.11 provides an upper bound on $\mathrm{KL}(P_{\boldsymbol{u}} \,\|\, Q_d)$ which is useful when $\sigma$ is large. Note that the KL divergence upper bounds the TV distance through Pinsker's or the Bretagnolle–Huber inequality (see e.g., [14]).

**Lemma B.10** (TV upper bound). *For any $\gamma > 1$ and $\sigma > 0$ such that $\sigma \geq 2/\gamma$, the following holds for the $\sigma$-smoothed discrete Gaussian $A_\gamma^\sigma$ and $Q = \mathcal{N}(0, 1/(2\pi))$.*

$$\mathrm{TV}(A_\gamma^\sigma, Q) \lesssim e^{-\pi s^2} \ ,$$

*where $s = \gamma\sigma/\sqrt{1 + \sigma^2}$.*

*Proof.* By the $L^1$-characterization of the TV distance,

$$\mathrm{TV}(A_\gamma^\sigma, Q) = \frac{1}{2} \int |A_\gamma^\sigma(x) - Q(x)| dx = \frac{1}{2} \int Q(x) |T_\gamma^\sigma(x) - 1| dx \ ,$$

where the likelihood ratio $T_\gamma^\sigma$ is given by (see Eq.(5))

$$
\begin{aligned}
T_\gamma^\sigma(x) &= \frac{\sqrt{1 + \sigma^2}}{\sigma \rho((1/\gamma)\mathbb{Z})} \sum_{k \in \mathbb{Z}} \rho_\sigma(x - \sqrt{1 + \sigma^2} k/\gamma) \\
&= \frac{\sqrt{1 + \sigma^2}}{\sigma \rho((1/\gamma)\mathbb{Z})} \sum_{k \in \mathbb{Z}} \rho_{\gamma\sigma/\sqrt{1+\sigma^2}}(\gamma x/\sqrt{1 + \sigma^2} - k) \ .
\end{aligned}
$$

The likelihood ratio is also a re-scaled version of the periodic Gaussian density since

$$\Psi_{\mathbb{Z}, s}(t) = \frac{\rho_s(\mathbb{Z} - t)}{s} = \frac{1}{s} \sum_{k \in \mathbb{Z}} \rho_s(k - t) \ . \tag{20}$$

Plugging in $s = \gamma\sigma/\sqrt{1 + \sigma^2} = \gamma/\sqrt{(1/\sigma^2) + 1}$ to Eq.(20), we have

$$T_\gamma^\sigma(x) = \frac{\gamma}{\rho((1/\gamma)\mathbb{Z})} \cdot \Psi_{\mathbb{Z}, s}(\gamma x/\sqrt{1 + \sigma^2}) \ .$$

The assumption $\sigma \geq 2/\gamma$ implies that $s \geq 1$. Thus, by Lemma B.8

$$
\begin{aligned}
|T_\gamma^\sigma(x) - 1| &\leq \frac{\gamma}{\rho((1/\gamma)\mathbb{Z})} \left| \Psi_{\mathbb{Z}, s}(\gamma x/\sqrt{1 + \sigma^2}) - 1 \right| + \left| \frac{\gamma}{\rho((1/\gamma)\mathbb{Z})} - 1 \right| \\
&\leq \frac{\gamma}{\rho((1/\gamma)\mathbb{Z})} \cdot 2(1 + 1/(\pi s)) e^{-\pi s^2} + \left| \frac{\gamma}{\rho((1/\gamma)\mathbb{Z})} - 1 \right| \\
&\leq \frac{\gamma}{\rho((1/\gamma)\mathbb{Z})} \cdot 3 e^{-\pi s^2} + \left| \frac{\gamma}{\rho((1/\gamma)\mathbb{Z})} - 1 \right| \ . 
\end{aligned}
\tag{21}
$$

Since $\rho((1/\gamma)\mathbb{Z}) = \rho_\gamma(\mathbb{Z}) = \gamma \Psi_{\mathbb{Z}, \gamma}(0)$ and $\gamma > 1$, by Lemma B.8 applied to $\Psi_{\mathbb{Z}, \gamma}$,

$$\left| \frac{\rho((1/\gamma)\mathbb{Z})}{\gamma} - 1 \right| \leq 2(1 + 1/(\pi\gamma)) e^{-\pi\gamma^2} < 3 e^{-\pi\gamma^2} < 1/4 \ .$$

We may thus write $\rho((1/\gamma)\mathbb{Z})/\gamma = 1 + \varepsilon$ for some $\varepsilon \in \mathbb{R}$ such that $|\varepsilon| \leq 3 e^{-\pi\gamma^2} < 1/4$. Then,

$$\left| \frac{\gamma}{\rho((1/\gamma)\mathbb{Z})} - 1 \right| = \left| \frac{1}{1 + \varepsilon} - 1 \right| = \left| \frac{\varepsilon}{1 + \varepsilon} \right| < (4/3)\varepsilon \ .$$

Applying the above inequalities to Eq.(21), we have that for any $x \in \mathbb{R}$,

$$\begin{aligned}
|T_\gamma^\sigma(x) - 1| &\leq \frac{\gamma}{\rho((1/\gamma)\mathbb{Z})} \cdot 3e^{-\pi s^2} + \left| \frac{\gamma}{\rho((1/\gamma)\mathbb{Z})} - 1 \right| \\
&< (1 + 3e^{-\pi\gamma^2}) \cdot 3e^{-\pi s^2} + 4e^{-\pi\gamma^2} \\
&\leq 4(e^{-\pi s^2} + e^{-\pi\gamma^2}) .
\end{aligned}$$

Since $0 < s < \gamma$, it follows that

$$\mathrm{TV}(A_\gamma^\sigma, Q) \leq 4(e^{-\pi s^2} + e^{-\pi\gamma^2}) \leq 8e^{-\pi s^2} .$$

$\square$

**Lemma B.11** (KL upper bound). *For any $\gamma > 0$ and $\sigma > 0$, the following holds for the $\sigma$-smoothed discrete Gaussian $A_\gamma^\sigma$ and $Q = \mathcal{N}(0, 1/(2\pi))$.*

$$\mathrm{KL}(A_\gamma^\sigma || Q) \leq \log(\sqrt{1+\sigma^2}/\sigma) \leq \frac{1}{2\sigma^2} .$$

*Expressed in terms of the time with respect to the OU process $e^{-t} = 1/\sqrt{1+\sigma^2}$, for any $t > 0$*

$$\mathrm{KL}(A_\gamma^\sigma \| Q) \leq \frac{e^{-2t}}{2(1 - e^{-2t})} .$$

*Proof.* By Jensen's inequality and the fact that for any $x \in \mathbb{R}$, $T_\gamma^\sigma(x) \leq T_\gamma^\sigma(0)$.

$$\begin{aligned}
\mathrm{KL}(A_\gamma^\sigma || Q) = \mathbb{E}_{x \sim A_\gamma^\sigma}[\log T_\gamma^\sigma(x)] &\leq \log \mathbb{E}_{x \sim A_\gamma^\sigma} T_\gamma^\sigma(x) \\
&\leq \log T_\gamma^\sigma(0) \\
&= \log \frac{\sqrt{1+\sigma^2}}{\sigma \rho_\gamma(\mathbb{Z})} \cdot \rho_s(\mathbb{Z}) ,
\end{aligned}$$

where $s = \gamma\sigma/\sqrt{1+\sigma^2} < \gamma$.

If $0 \leq s_1 \leq s_2$, then $\rho_{s_1}(\mathbb{Z}) \leq \rho_{s_2}(\mathbb{Z})$. Hence, $\rho_s(\mathbb{Z})/\rho_\gamma(\mathbb{Z}) \leq 1$. Using the fact that $\log(1+a) \leq a$ for any $a > -1$, we have

$$\mathrm{KL}(A_\gamma^\sigma \| Q) \leq \log(\sqrt{1+\sigma^2}/\sigma) = (1/2)\log(1 + 1/\sigma^2) \leq 1/(2\sigma^2) .$$

The second part of the lemma follows straightforwardly from the relation $\sigma/\sqrt{1+\sigma^2} = \sqrt{1 - e^{-2t}}$. Using the fact that $\log(1 - a) \geq -a/(1-a)$ for any $a < 1$, for any $t > 0$ we have

$$\mathrm{KL}(A_\gamma^\sigma \| Q) \leq \log(1/\sqrt{1 - e^{-2t}}) = -(1/2)\log(1 - e^{-2t}) \leq \frac{e^{-2t}}{2(1 - e^{-2t})} .$$

$\square$

## C   Proofs for Section 4

### C.1   Sample complexity of score estimation

We show that score estimation reduces to parameter estimation for Gaussian pancakes. Given that the sample complexity of parameter estimation for Gaussian pancakes is polynomial in the relevant problem parameters (Theorem 4.2), our reduction implies that the sample complexity of score estimation is polynomial as well.

**Lemma C.1** (Lemma 4.1 restated). *For any $\gamma > 1, \sigma > 0$, let $P_{\boldsymbol{u}}$ be the $(\gamma, \sigma)$-Gaussian pancakes distribution with secret direction $\boldsymbol{u} \in \mathbb{S}^{d-1}$. Given any $\eta \in (0, 1)$ and $\hat{\boldsymbol{u}} \in \mathbb{S}^{d-1}$ such that $1 - \langle \hat{\boldsymbol{u}}, \boldsymbol{u} \rangle^2 \leq \eta^2$, the score estimate $\hat{s}(\boldsymbol{x}) = -2\pi\boldsymbol{x} + \nabla \log T_\gamma^\sigma(\langle \boldsymbol{x}, \hat{\boldsymbol{u}} \rangle)$ satisfies*

$$\mathbb{E}_{\boldsymbol{x} \sim P_{\boldsymbol{u}}} \|\hat{s}(\boldsymbol{x}) - s(\boldsymbol{x})\|^2 \lesssim \max(1, 1/\sigma^8) \cdot \eta^2 d ,$$

*where $s(\boldsymbol{x}) = -2\pi\boldsymbol{x} + \nabla \log T_\gamma^\sigma(\langle \boldsymbol{x}, \boldsymbol{u} \rangle)$ is the score function of $P_{\boldsymbol{u}}$.*

*Proof.* For simplicity, we omit super and subscripts of the likelihood ratio $T_\gamma^\sigma$ and simply denote it by $T$. Let $\hat{u}$ be an estimate satisfying $1 - \langle \hat{u}, u \rangle^2 \le \eta^2$ and denote $\hat{u} = \langle \hat{u}, u \rangle u + w$. Note that $w \in \mathbb{R}^d$ is orthogonal to $u$ and $\|w\|^2 = 1 - \langle \hat{u}, u \rangle^2 \le \eta^2$. Then, we have

$$s(\boldsymbol{x}) - \hat{s}(\boldsymbol{x}) = \frac{T'(\langle \boldsymbol{x}, \boldsymbol{u} \rangle)}{T(\langle \boldsymbol{x}, \boldsymbol{u} \rangle)} \boldsymbol{u} - \frac{T'(\langle \boldsymbol{x}, \hat{\boldsymbol{u}} \rangle)}{T(\langle \boldsymbol{x}, \hat{\boldsymbol{u}} \rangle)} \hat{\boldsymbol{u}}$$

$$= \left( \frac{T'(\langle \boldsymbol{x}, \boldsymbol{u} \rangle)}{T(\langle \boldsymbol{x}, \boldsymbol{u} \rangle)} - \frac{T'(\langle \boldsymbol{x}, \hat{\boldsymbol{u}} \rangle)}{T(\langle \boldsymbol{x}, \hat{\boldsymbol{u}} \rangle)} \langle \boldsymbol{u}, \hat{\boldsymbol{u}} \rangle \right) \boldsymbol{u} - \frac{T'(\langle \boldsymbol{x}, \hat{\boldsymbol{u}} \rangle)}{T(\langle \boldsymbol{x}, \hat{\boldsymbol{u}} \rangle)} \boldsymbol{w}$$

By the triangle inequality and the fact that $(a+b)^2 \le 2a^2 + 2b^2$ for any $a, b \in \mathbb{R}$, we have

$$\|s(\boldsymbol{x}) - \hat{s}(\boldsymbol{x})\|^2 \le 2 \left( \frac{T'(\langle \boldsymbol{x}, \boldsymbol{u} \rangle)}{T(\langle \boldsymbol{x}, \boldsymbol{u} \rangle)} - \frac{T'(\langle \boldsymbol{x}, \hat{\boldsymbol{u}} \rangle)}{T(\langle \boldsymbol{x}, \hat{\boldsymbol{u}} \rangle)} \langle \boldsymbol{u}, \hat{\boldsymbol{u}} \rangle \right)^2 + 2 \left( \frac{T'(\langle \boldsymbol{x}, \hat{\boldsymbol{u}} \rangle)}{T(\langle \boldsymbol{x}, \hat{\boldsymbol{u}} \rangle)} \right)^2 \eta^2$$

$$\le 4 \left( \frac{T'(\langle \boldsymbol{x}, \boldsymbol{u} \rangle)}{T(\langle \boldsymbol{x}, \boldsymbol{u} \rangle)} - \frac{T'(\langle \boldsymbol{x}, \hat{\boldsymbol{u}} \rangle)}{T(\langle \boldsymbol{x}, \hat{\boldsymbol{u}} \rangle)} \right)^2 + 4 \left( \frac{T'(\langle \boldsymbol{x}, \hat{\boldsymbol{u}} \rangle)}{T(\langle \boldsymbol{x}, \hat{\boldsymbol{u}} \rangle)} \right)^2 (1 - |\langle \boldsymbol{u}, \hat{\boldsymbol{u}} \rangle|)^2 + 2 \left( \frac{T'(\langle \boldsymbol{x}, \hat{\boldsymbol{u}} \rangle)}{T(\langle \boldsymbol{x}, \hat{\boldsymbol{u}} \rangle)} \right)^2 \eta^2$$

$$\lesssim \left( \frac{T'(\langle \boldsymbol{x}, \boldsymbol{u} \rangle)}{T(\langle \boldsymbol{x}, \boldsymbol{u} \rangle)} - \frac{T'(\langle \boldsymbol{x}, \hat{\boldsymbol{u}} \rangle)}{T(\langle \boldsymbol{x}, \hat{\boldsymbol{u}} \rangle)} \right)^2 + \left( \frac{T'(\langle \boldsymbol{x}, \hat{\boldsymbol{u}} \rangle)}{T(\langle \boldsymbol{x}, \hat{\boldsymbol{u}} \rangle)} \right)^2 \eta^2 \tag{22}$$

By Lemma B.5 and the fact that $(1 + \sigma^2)/\sigma^2 \le 2\max(1, 1/\sigma^2)$, we know that the Lipschitz constant $L$ of $T'/T$ satisfies $L \lesssim \max(1, 1/\sigma^4)$. Furthermore, by Eq. (13) in Lemma B.5, we have $\|T'/T\|_\infty \lesssim \max(1, 1/\sigma^2)$. Applying these upper bounds to Eq.(22),

$$\|s(\boldsymbol{x}) - \hat{s}(\boldsymbol{x})\|^2 \lesssim \left( \frac{T'(\langle \boldsymbol{x}, \boldsymbol{u} \rangle)}{T(\langle \boldsymbol{x}, \boldsymbol{u} \rangle)} - \frac{T'(\langle \boldsymbol{x}, \hat{\boldsymbol{u}} \rangle)}{T(\langle \boldsymbol{x}, \hat{\boldsymbol{u}} \rangle)} \right)^2 + \left( \frac{T'(\langle \boldsymbol{x}, \hat{\boldsymbol{u}} \rangle)}{T(\langle \boldsymbol{x}, \hat{\boldsymbol{u}} \rangle)} \right)^2 \eta^2$$

$$\lesssim \max(1, 1/\sigma^8)(\langle \boldsymbol{x}, \boldsymbol{u} - \hat{\boldsymbol{u}} \rangle^2 + \eta^2)$$

$$\le \max(1, 1/\sigma^8)(\|\boldsymbol{x}\|^2 \|\boldsymbol{u} - \hat{\boldsymbol{u}}\|^2 + \eta^2)$$

$$\le \max(1, 1/\sigma^8) \cdot \eta^2 (\|\boldsymbol{x}\|^2 + 1) .$$

Since $\mathbb{E}_{\boldsymbol{x} \sim P_{\boldsymbol{u}}} \|\boldsymbol{x}\|^2 \le d$ by Lemma B.2, it follows that $\mathbb{E}_{\boldsymbol{x} \sim P_{\boldsymbol{u}}} \|s(\boldsymbol{x}) - \hat{s}(\boldsymbol{x})\|^2 \lesssim \max(1, 1/\sigma^8) \cdot \eta^2 d$ . $\qquad \square$

## C.2 Sample complexity of parameter estimation

For parameter estimation, we design a *contrast function* $g : \mathbb{R} \to \mathbb{R}$ such that the (population) functionals $G$ and $E$, defined below, are monotonic. For any $\gamma > 0$, $\boldsymbol{u} \in \mathbb{S}^{d-1}$, and $(\gamma, 0)$-Gaussian pancakes $P_{\boldsymbol{u}}$, we define

$$G(\sigma) := \mathbb{E}_{x \sim A_\gamma^\sigma}[g(x)] \tag{23}$$

$$E(\boldsymbol{v}) := \mathbb{E}_{\boldsymbol{x} \sim P_{\boldsymbol{u}}}[g(\langle \boldsymbol{x}, \boldsymbol{v} \rangle)] . \tag{24}$$

Note that $E(\boldsymbol{v}) = G(\sigma)$, where $\sigma^2 = (1 - \langle \boldsymbol{u}, \boldsymbol{v} \rangle^2)/\langle \boldsymbol{u}, \boldsymbol{v} \rangle^2$. We choose $g$ so that $G(\sigma)$ is decreasing in $\sigma$ and $E(\boldsymbol{v})$ is increasing in $\langle \boldsymbol{u}, \boldsymbol{v} \rangle^2$. We use $g = T_\gamma^\beta$ for some appropriately chosen $\beta > 0$. The monotonicity property of $T_\gamma^\beta$ is shown in Lemma C.3. Thus, given two candidate directions $\boldsymbol{v}_1, \boldsymbol{v}_2$, if $E(\boldsymbol{v}_1) \ge E(\boldsymbol{v}_2)$, then $\langle \boldsymbol{u}, \boldsymbol{v}_1 \rangle^2 \ge \langle \boldsymbol{u}, \boldsymbol{v}_2 \rangle^2$. This suggests the projection pursuit-based estimator $\hat{\boldsymbol{u}} = \arg\max_{\boldsymbol{v} \in \mathcal{C}} \hat{E}(\boldsymbol{v})$, where $\mathcal{C}$ is an $\eta$-net of the parameter space $\mathbb{S}^{d-1}$ and $\hat{E}(\boldsymbol{v}) = (1/n) \sum_{i=1}^n T_\gamma^\beta(\langle \boldsymbol{x}_i, \boldsymbol{v} \rangle)$ is the empirical version of $E$. The main theorem of this section (Theorem 4.2) shows that $n = \text{poly}(d, \gamma, 1/\eta)$ samples is sufficient for achieving $L^2$-error $\eta$ using the estimator $\hat{\boldsymbol{u}}$. We start with a useful fact about $\sigma$-smoothed likelihood ratios $T_\gamma^\sigma$.

**Claim C.2.** *For any $\beta > 0$, $\sigma \ge 0$, and $x \in \mathbb{R}$,*

$$\mathbb{E}_{z \sim Q}\left[ T_\gamma^\beta \left( \frac{1}{\sqrt{1 + \sigma^2}} x + \frac{\sigma}{\sqrt{1 + \sigma^2}} z \right) \right] = T_\gamma^s(x) ,$$

where $s = \sqrt{(1+\beta^2)(1+\sigma^2)-1}$ and $T_\gamma^\sigma$ denotes the likelihood ratio of the $\sigma$-smoothed discrete Gaussian $A_\gamma^\sigma$ on $(1/\gamma)\mathbb{Z}$ with respect to $Q = \mathcal{N}(0, 1/(2\pi))$ (see Eq. (5)).

*Proof.* Let $\tilde{x} = x/\sqrt{1+\sigma^2}$, $\tilde{\sigma} = \sigma/\sqrt{1+\sigma^2}$, and $c_k(\tilde{x}) = \tilde{x} - \sqrt{1+\beta^2}k/\gamma$. Then,

$$\mathbb{E}_Q\big[T_\gamma^\beta\big(\tilde{x} + \tilde{\sigma}z\big)\big] = \frac{\sqrt{1+\beta^2}}{\beta\rho((1/\gamma)\mathbb{Z})} \int \rho(z) \sum_{k\in\mathbb{Z}} \rho_\beta\bigg(\tilde{x} + \tilde{\sigma}z - \sqrt{1+\beta^2}k/\gamma\bigg)dz$$

$$= \frac{\sqrt{1+\beta^2}}{\beta\rho((1/\gamma)\mathbb{Z})} \sum_{k\in\mathbb{Z}} \int \rho_{\beta/\sqrt{\beta^2+\tilde{\sigma}^2}}(z - c_k(\tilde{x}))dz \cdot \rho_{\sqrt{\beta^2+\tilde{\sigma}^2}}(c_k(\tilde{x}))$$

$$= \frac{\sqrt{1+\beta^2}}{\sqrt{\beta^2+\tilde{\sigma}^2} \cdot \rho((1/\gamma)\mathbb{Z})} \sum_{k\in\mathbb{Z}} \rho_{\sqrt{\beta^2+\tilde{\sigma}^2}}(\tilde{x} - \sqrt{1+\beta^2}k/\gamma) .$$

Plugging in $\tilde{x} = x/\sqrt{1+\sigma^2}$ and $\tilde{\sigma} = \sigma/\sqrt{1+\sigma^2}$ gives us

$$\mathbb{E}_Q\bigg[T_\gamma^\beta\bigg(\frac{1}{\sqrt{1+\sigma^2}}x + \frac{\sigma}{\sqrt{1+\sigma^2}}z\bigg)\bigg] = \frac{\sqrt{1+s^2}}{s\rho((1/\gamma)\mathbb{Z})} \sum_{k\in\mathbb{Z}} \rho_s(x - \sqrt{1+s^2}k/\gamma) = T_\gamma^s(x) ,$$

where $s = \sqrt{(1+\beta^2)(1+\sigma^2)-1}$. $\qquad\square$

**Lemma C.3** (Monotonicity). *Given any $\gamma > 0$ and $\beta > 0$, define $G(\sigma) := \mathbb{E}_{x\sim A_\gamma^\sigma}[T_\gamma^\beta(x)]$, where $T_\gamma^\sigma$ denotes the likelihood ratio of the $\sigma$-smoothed discrete Gaussian $A_\gamma^\sigma$ on $(1/\gamma)\mathbb{Z}$ with respect to $\mathcal{N}(0, 1/(2\pi))$. Then, for any $\sigma > 0$, we have $G'(\sigma) < 0$.*

*Proof.* Let $T_\gamma^0(x) = \sum_{k=0}^\infty \alpha_{2k}h_{2k}(x)$ be the (formal) Hermite expansion of $T_\gamma^0(x)$, where $(h_k)_{k\in\mathbb{N}}$ form an orthonormal sequence with respect to $\langle\cdot,\cdot\rangle_Q$. By Claim C.2, for any $\sigma \geq 0$

$$T_\gamma^\sigma(x) = \sum_{k=0}^\infty \alpha_{2k}\bigg(\frac{1}{1+\sigma^2}\bigg)^k h_{2k}(x) . \tag{25}$$

Using the orthonormality of $(h_k)$, for any $\sigma > 0$

$$G(\sigma) = \mathbb{E}_{x\sim A_\gamma^\sigma}[T_\gamma^\beta(x)] = \langle T_\gamma^\beta, T_\gamma^\sigma\rangle_Q = \sum_{k=0}^\infty \alpha_{2k}^2 \cdot \frac{1}{(1+\beta^2)^k(1+\sigma^2)^k}$$

$$G'(\sigma) = -\sum_{k\geq 1} \alpha_{2k}^2 \cdot \frac{1}{(1+\beta^2)^k} \cdot \frac{2k\sigma}{(1+\sigma^2)^{k+1}} < 0 .$$

$\qquad\square$

**Lemma C.4** (Non-trivial Hermite coefficient). *Let $(h_k)$ be the normalized Hermite polynomials with respect to $\langle\cdot,\cdot\rangle_Q$. For any $\gamma > 1$ such that $\pi\gamma^2 \in \mathbb{N}$ and $\ell \in \mathbb{N}$, it holds that*

$$\big|\mathbb{E}_{x\sim A_\gamma} h_{2\pi\ell^2\gamma^2}(x)\big| \geq \frac{1}{\sqrt{2\pi\ell\gamma}} ,$$

*where $A_\gamma$ is the discrete Gaussian on $(1/\gamma)\mathbb{Z}$.*

*Proof.* Let $(H_k)$ be the unnormalized Hermite polynomials defined by

$$H_k(x)\rho(x) = (-1)^k \frac{d^k}{dx^k}\rho(x) .$$

Using the relation between the Fourier transform and differentiation, we have

$$\mathcal{F}\{H_k(x)\rho(x)\} = \mathcal{F}\bigg\{(-1)^k \frac{d^k}{dx^k}\rho(x)\bigg\} = (-2\pi iy)^k \rho(y) .$$

Let $h_k = c_k H_k$ be the normalized Hermite polynomials, where $c_k = \sqrt{(2\pi)^k k!}$ (see e.g., [64, Chapter 4.2.1]). By the Poisson summation formula (Lemma 2.1), we have

$$\mathbb{E}_{x \sim A_\gamma} h_k(x) = \frac{1}{\rho((1/\gamma)\mathbb{Z})} \sum_{x \in (1/\gamma)\mathbb{Z}} c_k H_k(x)\rho(x)$$

$$= \frac{\gamma c_k}{\rho((1/\gamma)\mathbb{Z})} \sum_{y \in \gamma\mathbb{Z}} (-2\pi i y)^k \rho(y) .$$

We now analyze the maximum among the terms inside the sum. Let $f(y) = y^k \rho(y)$. Note that

$$f(y) = y^k \rho(y) = (2\pi)^k \exp(-\pi y^2 + k \log y) .$$

The exponent in the above expression is maximized at $2\pi(y^*)^2 = k$. This maximum is indeed achieved in the sum since we can choose $y^* = \ell\gamma$, which is permissible given the assumption $\pi\gamma^2 \in \mathbb{N}$. Hence, the maximum value of $f(y)$ is

$$f(y^*) = \exp(-k/2 + (k/2)\log(k/2\pi)) = (k/2e\pi)^{k/2} .$$

Plugging in the value $c_k = 1/\sqrt{(2\pi)^k k!}$ and using the Stirling lower bound $k! \geq \sqrt{2\pi k}(k/e)^k$ for all $k \in \mathbb{N}$,

$$\left| \mathbb{E}_{x \sim A_\gamma} h_k(x) \right| \geq \frac{2\gamma}{\rho((1/\gamma)\mathbb{Z})} c_k (2\pi)^k (k/2e\pi)^{k/2}$$

$$= \frac{2\gamma}{\rho((1/\gamma)\mathbb{Z})} c_k (2\pi k/e)^{k/2}$$

$$= \frac{2\gamma}{\rho((1/\gamma)\mathbb{Z})} \cdot \frac{(k/e)^{k/2}}{\sqrt{k!}}$$

$$\geq \frac{2\gamma}{\rho((1/\gamma)\mathbb{Z})} \cdot \frac{1}{(2\pi k)^{1/4}} .$$

In addition, we have that

$$\rho((1/\gamma)\mathbb{Z}) = \gamma\rho(\gamma\mathbb{Z}) \leq \gamma\rho(\mathbb{Z}) \leq 2\gamma .$$

Plugging in $k = 2\pi\ell^2\gamma^2$, we therefore have

$$\left| \mathbb{E}_{x \sim A_\gamma} h_k(x) \right| \geq \frac{1}{\sqrt{2\pi\ell\gamma}} .$$

$\square$

**Corollary C.5** (Non-trivial Hermite coefficient, rounded degree). *Let $(h_k)$ be the normalized Hermite polynomials with respect to $\langle \cdot, \cdot \rangle_Q$. For any $\gamma > 1$, $\ell \in \mathbb{N}$, and $k = 2\lfloor \ell^2 \pi \gamma^2 \rfloor$, it holds that*

$$\left| \mathbb{E}_{x \sim A_\gamma} h_k(x) \right| \geq \frac{1}{e\sqrt{2\pi\ell\gamma}} ,$$

*where $A_\gamma$ is the discrete Gaussian on $(1/\gamma)\mathbb{Z}$.*

*Proof.* Let $\ell^2 \pi \gamma^2 - \lfloor \ell^2 \pi \gamma^2 \rfloor = \alpha$. Then, $\alpha \in [0, 1)$ and $k = \ell^2(2\pi\gamma^2) - 2\alpha$. Similar to the proof of Lemma C.4, we apply the Poisson summation formula (Lemma 2.1) as follows.

$$\mathbb{E}_{x \sim A_\gamma} h_k(x) = \frac{1}{\rho((1/\gamma)\mathbb{Z})} \sum_{x \in (1/\gamma)\mathbb{Z}} h_k(x)\rho(x)$$

$$= \frac{\gamma}{\rho((1/\gamma)\mathbb{Z})} \sum_{y \in \gamma\mathbb{Z}} c_k(-2\pi i y)^k \rho(y) .$$

As previously shown in Lemma C.4, the function $f(y) = y^k \rho(y)$ achieves its maximum at $(y^*)^2 = k/2\pi = \ell^2 \gamma^2 - \alpha/\pi$. The issue now is that $y^*$ is not necessarily contained in $\gamma \mathbb{Z}$. However, we show that "rounding up" $y^*$ to $s\gamma$ is sufficient to establish a non-trivial lower bound. Taking the ratio of $f(y^*)$ and $f(s\gamma)$,

$$\log f(s\gamma)/f(y^*) = k \log \left( \frac{\ell \gamma}{y^*} \right) - \pi(\ell^2 \gamma^2 - (y^*)^2) \geq -\alpha > -1 \ .$$

Hence, $f(s\gamma) > f(y^*)/e$. Combining this observation and the proof of Lemma C.4 leads to the conclusion. $\qquad \square$

**Theorem C.6** (Theorem 4.2 restated). *For any constant $C > 0$, given $\gamma(d) > 1, \sigma(d) > 0$ such that $\gamma\sigma < C$, estimation error parameter $\eta > 0$, and $\delta \in (0,1)$, there exists a brute-force search estimator $\hat{u} : \mathbb{R}^{d \times n} \to \mathbb{S}^{d-1}$ that uses $n = \mathrm{poly}(d, \gamma, 1/\eta, 1/\delta)$ samples and achieves $\|\hat{u}(x_1, \ldots, x_n) - u\|^2 \leq \eta^2$ with probability at least $1 - \delta$ over i.i.d. samples $x_1, \ldots, x_n \sim P_u$, where $P_u$ is the $(\gamma, \sigma)$-Gaussian pancakes distribution with secret direction $u \in \mathbb{S}^{d-1}$.*

*Proof.* Without loss of generality, we assume $\eta \leq 1/\gamma$. If the given error parameter $\eta$ is larger than $1/\gamma$, we set $\eta = 1/\gamma$. Let $\mathcal{C}$ be any $\eta$-net of $\mathbb{S}^{d-1}$. Our brute-force search estimator is

$$\hat{u} = \arg\max_{v \in \mathcal{C}} \hat{E}(v) \ , \qquad \text{where } \hat{E}(v) = \frac{1}{n} \sum_{i=1}^{n} T_\gamma^\beta(\langle x_i, v \rangle) \ .$$

We choose $\beta = 1/\sqrt{\pi}\gamma$ as the contrast function parameter for reasons explained later. The population limit of $\hat{E}$ is $E$ (Eq.(24)), and the monotonicity of $E$ with respect to $\langle u, v \rangle^2$ (Lemma C.3) implies that $u = \arg\max_{\mathbb{S}^{d-1}} E(v)$ in the infinite-sample limit. For $x \sim P_u$, the distribution of $\langle x, v \rangle$ is $A_\gamma^\xi$, where $\xi^2 = (1 + \sigma^2)/\langle u, v \rangle^2 - 1$. Let $v_1, v_2 \in \mathbb{S}^{d-1}$ be such that $\langle u, v_1 \rangle^2 - \langle u, v_2 \rangle^2 = \varepsilon^2$. Let $\xi_1^2 = (1 + \sigma^2)/\langle u, v_1 \rangle^2 - 1$ and $\xi_2^2 = (1 + \sigma^2)/\langle u, v_2 \rangle^2 - 1$. Notice that

$$\frac{1}{1 + \xi_1^2} - \frac{1}{1 + \xi_2^2} = \frac{\varepsilon^2}{1 + \sigma^2} \ .$$

For $\varepsilon$-far pairs $v_1, v_2$ such that $\xi_1$ is sufficiently small, we establish a lower bound on $E(v_1) - E(v_2)$ in terms of $\varepsilon$. This implies that if $E(v_1)$ and $E(v_2)$ are close, then $\langle u, v_1 \rangle^2$ and $\langle u, v_2 \rangle^2$ are also close.

By Claim C.2, for any $v \in \mathbb{S}^{d-1}$, we have

$$E(v) = \langle T_\gamma^\xi, T_\gamma^\beta \rangle_Q = \sum_{k=0}^{\infty} \alpha_{2k}^2 \cdot \frac{1}{(1 + \beta^2)^k (1 + \xi^2)^k} \ .$$

Since all the terms in the series are non-negative and monotonically decreasing in $\xi$, for any $k \in \mathbb{N}$ and $v_1, v_2 \in \mathbb{S}^{d-1}$ such that $\xi_1 \leq \xi_2$, we have

$$
\begin{aligned}
E(v_1) - E(v_2) &\geq \frac{\alpha_{2k}^2}{(1 + \beta^2)^k} \left( \left( \frac{1}{1 + \xi_1^2} \right)^k - \left( \frac{1}{1 + \xi_2^2} \right)^k \right) \\
&= \frac{\alpha_{2k}^2}{(1 + \beta^2)^k (1 + \xi_1^2)^k} \left( 1 - \frac{1 + \xi_1^2}{1 + \xi_2^2} \right) \left( \left( \frac{1 + \xi_1^2}{1 + \xi_2^2} \right)^{k-1} + \left( \frac{1 + \xi_1^2}{1 + \xi_2^2} \right)^{k-2} + \cdots + 1 \right) \\
&\geq \frac{\alpha_{2k}^2}{(1 + \beta^2)^k (1 + \xi_1^2)^k} \left( \frac{\varepsilon^2 (1 + \xi_1^2)}{1 + \sigma^2} \right) \\
&\geq \frac{\alpha_{2k}^2}{(1 + \beta^2)^k (1 + \xi_1^2)^{k-1}} \left( \frac{\varepsilon^2}{1 + \sigma^2} \right)
\end{aligned}
$$

We choose $k = \lfloor \pi\gamma^2 \rfloor, \beta^2 = 1/k$. If $\xi_1^2 \leq C'/k$ for some constant $C' > 0$ (this will be justified later), then by Corollary C.5, we have that $\alpha_{2k}^2 \geq 1/(2\pi e^2 \gamma)$. Thus, using the fact that $1 + t \leq e^t$ for any $t \in \mathbb{R}$,

$$\frac{\alpha_{2k}^2}{(1 + \beta^2)^k (1 + \xi_1^2)^{k-1}} \left( \frac{\varepsilon^2}{1 + \sigma^2} \right) \geq \frac{\alpha_{2k}^2}{e^{C'+1}} \left( \frac{\varepsilon^2}{1 + \sigma^2} \right) \geq \frac{1}{2\pi e^{C'+3} \gamma} \left( \frac{\varepsilon^2}{1 + \sigma^2} \right) \ .$$

Since $\sigma^2 \le C^2/\gamma^2 < C^2$, it follows that

$$E(\boldsymbol{v}_1) - E(\boldsymbol{v}_2) > \frac{1}{2\pi(1+C^2)e^{C'+3}} \cdot \frac{\varepsilon^2}{\gamma} . \tag{26}$$

Taking the contrapositive, if $\boldsymbol{v}_1, \boldsymbol{v}_2 \in \mathbb{S}^{d-1}$ are such that $\xi_1 \le \xi_2$, $\xi_1^2 \le C'/k$ and $E(\boldsymbol{v}_1) - E(\boldsymbol{v}_2) \le \varepsilon^2/(2\pi(1+C^2)e^{C'+3}\gamma)$, then $\langle \boldsymbol{u}, \boldsymbol{v}_1 \rangle^2 - \langle \boldsymbol{u}, \boldsymbol{v}_2 \rangle^2 \le \varepsilon^2$.

Equipped with this result, we revisit our $\eta$-net $\mathcal{C}$ of $\mathbb{S}^{d-1}$. Let $\boldsymbol{v}^* = \arg\max_{\boldsymbol{v}\in\mathcal{C}} E(\boldsymbol{v})$ be the *population* maximizer of $E$ within $\mathcal{C}$. By the monotonicity of $E(\boldsymbol{v})$ with respect to $\langle \boldsymbol{u}, \boldsymbol{v} \rangle^2$, we have that $1 - \langle \boldsymbol{u}, \boldsymbol{v}^* \rangle^2 \le \eta^2/2$. Moreover, since by assumption $\sigma^2 \le C^2/\gamma^2$ and $\eta^2 \le 1/\gamma^2 \le 1$, the corresponding noise level $(\xi^*)^2 := (1+\sigma^2)/\langle \boldsymbol{u}, \boldsymbol{v}^* \rangle^2 - 1$ satisfies

$$(\xi^*)^2 \le \frac{1+\sigma^2}{1-\eta^2/2} - 1 = \frac{\sigma^2 + \eta^2/2}{1-\eta^2/2} \le 2\sigma^2 + \eta^2 \le (2C^2+1)/\gamma^2 . \tag{27}$$

Now consider $\hat{\boldsymbol{u}} = \arg\max_{\boldsymbol{v}\in\mathcal{C}} \hat{E}(\boldsymbol{v})$, the maximizer of the *empirical* objective. By the triangle inequality,

$$\begin{aligned}
E(\boldsymbol{v}^*) - E(\hat{\boldsymbol{u}}) &= (E(\boldsymbol{v}^*) - \hat{E}(\boldsymbol{v}^*)) + (\hat{E}(\boldsymbol{v}^*) - \hat{E}(\hat{\boldsymbol{u}})) + (\hat{E}(\hat{\boldsymbol{u}}) - E(\hat{\boldsymbol{u}})) \\
&\le |E(\boldsymbol{v}^*) - \hat{E}(\boldsymbol{v}^*)| + |\hat{E}(\hat{\boldsymbol{u}}) - E(\hat{\boldsymbol{u}})| .
\end{aligned} \tag{28}$$

We show that both terms in Eq.(28) concentrate. Recall that $\hat{E}(\boldsymbol{v}) = (1/n)\sum_{i=1}^{n} T_\gamma^\beta(\langle \boldsymbol{v}, \boldsymbol{x}_i \rangle)$ and that the function $T_\gamma^\beta$ satisfies (see proof of Lemma B.11)

$$T_\gamma^\beta(z) \le T_\gamma^\beta(0) = \frac{\sqrt{1+\beta^2}}{\beta\rho((1/\gamma)\mathbb{Z})} \cdot \rho((\sqrt{1+\beta^2}/\beta\gamma)\mathbb{Z}) \le \sqrt{1+\beta^2}/\beta .$$

Since $\beta^2 = 1/(\pi\gamma^2)$, we have $T_\gamma^\beta(z) \le \gamma\sqrt{2\pi}$ for any $z \in \mathbb{R}$. Thus, for any fixed $\boldsymbol{v} \in \mathbb{S}^{d-1}$, $\hat{E}(\boldsymbol{v})$ is a sum of i.i.d. bounded random variables $T_\gamma^\beta$. By Hoeffding's inequality, $n \lesssim (\gamma^3/\eta^4)\log(|\mathcal{C}|/\delta)$ samples are sufficient to guarantee that for all $\boldsymbol{v} \in \mathcal{C}$, $|E(\boldsymbol{v}) - \hat{E}(\boldsymbol{v})| \lesssim \eta^2/\gamma$ with probability at least $1-\delta$. From standard covering number bounds (e.g., [77, Corollary 4.2.13]), we have $|\mathcal{C}| \le (3/\eta)^d$. Hence, $n \lesssim (d\gamma^3/\eta^4)(\log(1/\eta) + \log(1/\delta))$ samples are sufficient for the desired level of concentration.

It follows that $E(\boldsymbol{v}^*) - E(\hat{\boldsymbol{u}}) \lesssim \eta^2/\gamma$ with probability $1-\delta$ over the randomness of $\hat{\boldsymbol{u}}$. Since $\boldsymbol{v}^*$ satisfies $(\xi^*)^2 \lesssim 1/\gamma^2$ (see Eq.(27)) and $\xi^* \le \xi$, where we denote $\xi^2 = (1+\sigma^2)/\langle \boldsymbol{u}, \hat{\boldsymbol{u}} \rangle^2 - 1$, the closeness of $E(\boldsymbol{v}^*)$ and $E(\hat{\boldsymbol{u}})$ implies $\langle \boldsymbol{u}, \boldsymbol{v}^* \rangle^2 - \langle \boldsymbol{u}, \hat{\boldsymbol{u}} \rangle^2 \le \eta^2$ due to Eq.(26). Hence,

$$\|\boldsymbol{u} - \hat{\boldsymbol{u}}\|^2 = 2(1 - \langle \boldsymbol{u}, \hat{\boldsymbol{u}} \rangle^2) \le 2\eta^2 + 2(1 - \langle \boldsymbol{u}, \boldsymbol{v}^* \rangle^2) \le 3\eta^2 .$$

$\square$

