# OpenReview forum: "Cryptographic Hardness of Score Estimation"
_NeurIPS.cc/2024/Conference — NeurIPS 2024 poster_

### Official Review · Reviewer_CMgH · 2024-07-07

**Soundness:** 3
**Presentation:** 2
**Contribution:** 2
**Rating:** 5
**Confidence:** 3

**Summary:**

This paper investigates the difficulty of distinguishing a Gaussian pancake distribution from a Gaussian distribution from the perspective of score estimation. The authors show that, by assuming the $L^2$-error of the score estimation is of the order $\log(d)$, there is a polynomial-time algorithm that solves the Gaussian pancake problem. Moreover, they show that for the cryptographically hard regime, there is a polynomial sample complexity $O(d)$ that ensures a small score estimation error for Gaussian pancake problems.

**Strengths:**

The authors introduce a new perspective on the Gaussian pancake problem by presenting the score estimation problem from diffusion models. Specifically, their Theorem 3.1 demonstrates that if the score is estimated accurately, then the Gaussian pancake problem can be solved in polynomial time. This perspective is novel for the Gaussian pancake problem.

**Weaknesses:**

One main selling point, which has been emphasized repeatedly in the paper and is also the title of this paper, is that the score estimation is cryptographically hard. This is confusing to me. In fact, as I understand it, the main theorem states that the Gaussian pancake problem can be solved under certain assumptions, while the hardness seems to state that the score estimation is hard.

If I understand correctly, the hardness arises from the assumption that $\gamma\sigma = O(1)$ in Theorem 4.2, which is known to be hard in existing literature, while the score estimation has a polynomial upper bound. However, I cannot see the hardness from Theorem 4.2. Even though the Gaussian pancake problem is hard, the score estimation is accurate. If we further combine Theorem 4.2 with Theorem 3.1, I guess the authors want to argue that if the score estimation is easy, then the Gaussian pancake problem can be solved. However, as it is already known that the Gaussian pancake problem is hard, the score estimation cannot be easy. But the score estimation error assumption in Theorem 3.1 does not match Theorem 4.2's conclusion. In fact, the $O(\log d)$ assumptions of Theorem 3.1 on the estimation error are smaller than the polynomial $O(d)$ upper bound in Theorem 4.2, which may make the story incomplete.

Please correct me if my understanding is incorrect.

**Questions:**

please solve my concern in the weaknesses part

**Limitations:**

Yes

---

> ### Author Rebuttal · Authors · 2024-07-31
>
> We thank the reviewer for clearly communicating points of confusion and giving us the opportunity to clarify our results. Before providing any clarifications, we would like to mention that the reviewer’s summary below accurately and succinctly captures the main contribution of our paper.
>
> > *I guess the authors want to argue that if the score estimation is easy, then the Gaussian pancake problem can be solved. However, as it is already known that the Gaussian pancake problem is hard, the score estimation cannot be easy.*
>
> We now address specific questions raised by the reviewer.
>
> > *… as I understand it, the main theorem states that the Gaussian pancake problem can be solved under certain assumptions, while the hardness seems to state that the score estimation is hard.*
>
> We show that the Gaussian pancakes problem can be solved *if* we can solve L2-accurate score estimation for Gaussian pancakes distributions. More precisely, if an **efficient** algorithm existed for estimating the scores of Gaussian pancakes, it would imply the existence of an **efficient** algorithm for solving the Gaussian pancakes problem. This, as the reviewer correctly observed, is a contradiction since the Gaussian pancakes problem is known to be computationally hard.
>
> The qualifier “efficient” is crucial here. Our score estimation algorithm for Gaussian pancakes (presented in Section 4) is **not** efficient; it involves a brute-force search over the d-dimensional unit sphere which requires exponential-in-d time. The polynomial quantity in Theorem 4.2 refers to the **number of samples**, not the **running time** of the estimator.
>
> The main purpose of our inefficient estimator in Section 4 is to highlight a *gap* between what’s statistically possible, with no limits on computation, and what’s computationally feasible in poly(d) time (hence the name, statistical-to-computational *gap*). If a statistical problem is impossible even with infinite computation, then any claim of its computational hardness is vacuous. Section 4 demonstrates that our computational hardness result from Section 3 is not vacuous, as L2-accurate score estimation for Gaussian pancakes can be achieved with exp(d) computation time (and poly(d) samples).
>
> > *But the score estimation error assumption in Theorem 3.1 does not match Theorem 4.2's conclusion. In fact, the $O(\log d)$ assumptions of Theorem 3.1 on the estimation error are smaller than the polynomial $O(d)$ upper bound in Theorem 4.2, which may make the story incomplete.*
>
> To provide context, the $\epsilon = O(1/\sqrt{\log d})$ assumption in Theorem 3.1 refers to an upper bound on the **L2 score estimation error** $\epsilon$. Meanwhile, the $n = \mathrm{poly}(d, \gamma, 1/\eta)$ in Theorem 4.2 refers to the **number of samples** sufficient for estimating the Gaussian pancakes secret direction up to L2-error $\eta$. The magnitudes of these two quantities ($\epsilon$ and $n$) are not meant to be directly compared.
>
> Instead, the two quantities are related as follows. For any score estimation algorithm (whether efficient or not) to satisfy the L2 score estimation error assumption $\epsilon = O(1/\sqrt{\log d})$ in Theorem 3.1, it is *statistically sufficient* to choose $n$ to be some large **polynomial in $d$**. In other words, there exists a score estimation algorithm that achieves the L2-error bound $\epsilon = O(1/\sqrt{\log d})$  using a polynomial number of samples. Our Theorem 4.2 provides a **computationally inefficient** score estimator with such guarantees.
>
> We hope this explanation resolves any confusion and respectfully ask if the reviewer would be willing to reconsider our rating in light of this clarification. Thank you again for your time and consideration.

---

### Official Review · Reviewer_q3zX · 2024-07-12

**Soundness:** 3
**Presentation:** 4
**Contribution:** 3
**Rating:** 7
**Confidence:** 2

**Summary:**

This paper shows that $L^2$-accurate score estimation, a crucial primitive in the theory of diffusion models and sampling, is computationally hard in the worst-case. The main theorem is a negative result that provides a statistical-computational gap: if computationally efficient $L^2$-accurate score estimation is possible, then one has a computationally efficient algorithm for solving the *Gaussian pancakes problem*, a hard instance under widely believed hardness assumptions from lattice-based cryptography. This negative result is particularly important in relation to Chen et al. (2023), as they showed that access to an $L^2$-accurate score estimation oracle (with some mild additional assumptions) admits an algorithm for sampling from any arbitrary distribution. The computational hardness of such an oracle suggests future directions for research in making stronger assumptions so such an oracle *is* possible (omitting the worst-case instance of Gaussian pancakes) or weaker criteria for understanding when a sample generated from the DDPM process is "good enough."

The main result, Theorem 3.1, is proven mainly by way of using Theorem 2 of Chen et al. (2023), which states that, for any distribution, the output of the DDPM algorithm provides a certificate for Gaussianity. Importantly, this DDPM algorithm assumes access to an $L^2$-accurate score estimation oracle, so by using this certificate of Gaussianity on the particular choice of the Gaussian pancake distribution, we can distinguish just by using the certificate of Gaussianity as a test statistic. Theorem 4.2 also gives the sample complexity of estimating the Gaussian pancake distribution through a brute force search over the possible hidden directions.

**Strengths:**

This paper is very well-written and clear, and its main result, Theorem 3.1, is an interesting addition to the literature on both score estimation in diffusion model theory and the literature on statistical-computational gaps. I am not not an expert in either of these fields, particularly not in diffusion model theory, but I was able to mostly follow along with the proof and main statements in the work. However, because of my lack of previous exposure, I would take my words with a grain of salt.

**Originality:** As far as I know, this work is the only one to evaluate the statistical-computational gap of the $L^2$-accurate score estimation oracle in DDPM. The proof's use of Gaussian pancakes as the hard distribution is certainly original and interesting, and I believe that the result itself is an original contribution to both the literature on diffusion models and the learning theory literature on proving computational hardness results for statistical problems.

**Quality:** The work proves a theorem, Theorem 3.1, that is well-motivated and has ample analysis and interpretation to supplement the main claim. As far as my understanding goes, the proofs seem to be correct, though I may have not understood some details in the introduction of the stochastic differential equation defining the diffusion model process.

**Clarity:** The paper is very well-written and clear. Although I am an outsider to the diffusion model theory literature, I was able to roughly follow along with the arguments and the main theorem seemed very well-motivated after Sections 1 and 2. In particular, the authors do a very good job in sketching the implications of their main result in Section 1.3, and I appreciated the clarity of that section for bringing to light the significance of the result.

**Significance:** I am an outsider to the field of diffusion models, so I cannot confidently speak to the significance, but the result does seem very well-motivated and important, as, to my understanding, the score estimation oracle is central to the diffusion model process. Outside of the diffusion model literature, I believe that the general technique of using the Gaussian panacakes distribution as a hard instance for a statistical-computational gap is worth highlighting as an important technique. For this alone, I believe this paper presents a valuable contribution.

**Weaknesses:**

The paper was well-written and I could not find any typographical or clarity issues. To the best of my checking, the theorem seemed correct, and its proof outline provided insight into the argument. I am an outsider to the literature, so I would not be confident in critiquing the paper in terms of weaknesses. As far as I can see, it the paper is clear, the theorem is well-motivated, and the argument is correct.

If anything, one might critique that perhaps the work would be more complete with a positive result towards a distribution-specific narrowing of DDPM that avoids the Gaussian pancakes distributions. However, I have no gauge on how hard/standard this might be in the literature, and it seems that $L^2$-accurate score estimation is central enough a primitive in DDPM that this negative result is important in its own right.

**Questions:**

I have a couple of main questions, but they may stem from my ignorance of the literature:

1. Because the Gaussian pancakes class is hard, I assume that there have been positive results in the light of this computationally hard instance in other problems that exclude this class. What are examples of such distribution-specific guarantees?
2. How does the "weaker criteria for evaluating sample quality" in Section 1.3 relate to the Theorem 4.2? Is there still a statistical-computational gap if we allowed "distinguish" to be a looser requirement?

**Limitations:**

The authors have addressed the limitations of the work in the NeurIPS paper checklist. Because this is mainly a theory paper, I do not see it as having other ethical conflicts.

---

> ### Author Rebuttal · Authors · 2024-08-02
>
> We thank the reviewer for the positive feedback and the detailed review of our paper. The reviewer's summary effectively and accurately captures the essence of our results. We address elements of the review below, which highlight the strengths of our paper and pose insightful questions that deserve further investigation.
>
> > *The paper is very well-written and clear … the main theorem seemed very well-motivated after Sections 1 and 2. In particular, the authors do a very good job in sketching the implications of their main result in Section 1.3, and I appreciated the clarity of that section for bringing to light the significance of the result.*
>
> We are grateful for the reviewer’s appreciation of the motivation behind our work and the clarity of our exposition.
>
> > *… the work would be more complete with a positive result towards a distribution-specific narrowing of DDPM that avoids the Gaussian pancakes distributions. However, I have no gauge on how hard/standard this might be in the literature, and it seems that L2-accurate score estimation is central enough a primitive in DDPM that this negative result is important in its own right.*
>
> We agree that pairing negative results with positive results might provide a more complete understanding of the computational landscape of L2-accurate score estimation. However, as the reviewer noted, we believe that our negative result is significant in its own right. Moreover, we think that positive results warrant separate and careful treatment.
>
> There are various ways of excluding Gaussian pancakes, each leading to an intriguing class of distributions with its own set of technical challenges. Specific examples will be provided when addressing the reviewer’s **Question 1**. However, it’s important to note that each of these distribution classes has been explored in separate papers, each involving non-trivial analyses. Thus, from the perspective of topic homogeneity, it is not clear which classes should be analyzed within this paper and which might be better suited for separate investigation.
>
> > *1. Because the Gaussian pancakes class is hard, I assume that there have been positive results in the light of this computationally hard instance in other problems that exclude this class. What are examples of such distribution-specific guarantees?*
>
> This is a great question. In the context of computationally efficient score estimation, recent works have analyzed mixtures of Gaussians with either a fixed number of components or "well-conditioned" components. In particular, Shah et al. [SCK23] demonstrated a score estimator for mixtures of spherical Gaussians with well-separated means, while Chen et al. [CKS24] provided a score estimator for $k$-mixtures of arbitrary Gaussians, with a running time that depends *exponentially* on $k$ and a certain "condition number" $\tau$ of the mixture components. Note that the running time of Chen et al.’s score estimator is polynomial in the data dimension $d$ if $k$ and $\tau$ are constants with respect to $d$. These distribution classes exclude Gaussian pancakes since the pancakes have degenerate covariances and the number of components $k$ grows with $d$ via the parameter $\gamma$.
>
> Another class of interesting distributions is motivated by non-Gaussian component analysis (NGCA). A prototypical distribution arising in NGCA consists of a low-dimensional non-Gaussian "signal" embedded in high-dimensional Gaussian noise. More precisely, there exists a hidden $k$-dimensional subspace $V$ such that the projection of the distribution onto $V$ is non-Gaussian, while the projection onto its orthogonal complement $V^\perp$ is Gaussian. In standard NGCA settings where both $k$ and the non-Gaussian signal distribution are kept fixed with respect to the dimension $d$, polynomial time estimators for the hidden subspace $V$ are known [TV18, GS19].
>
> Again, Gaussian pancakes distributions are excluded from this class since their non-Gaussian "signal" distribution depends on $d$ via the parameters $\gamma$ and $\sigma$. In standard NGCA the difficulty stems solely from the signal being embedded in higher dimensions. In Gaussian pancakes, the difficulty is twofold: not only is the signal embedded in higher dimensions, but the 1D discrete Gaussian with spacing $1/\gamma$ also becomes increasingly difficult to distinguish from the 1D standard Gaussian as $\gamma$ grows with $d$.
>
> > *2. How does the "weaker criteria for evaluating sample quality" in Section 1.3 relate to the Theorem 4.2? Is there still a statistical-computational gap if we allowed "distinguish" to be a looser requirement?*
>
> Another great question. We do not have clear answers, but it would depend on the specific weaker criterion used. For example, if the weaker criterion were "match the first two moments", then we do not anticipate any statistical-computational gap. An algorithm that simply outputs $\mathcal{N}(\hat{\mu}, \hat{\Sigma})$, where $\hat{\mu}, \hat{\Sigma}$ are the empirical mean and covariance, would suffice. Thus, for this nearly trivial criterion, we can bypass L2-accurate score estimation entirely and directly satisfy the weak criterion.
>
> We believe that understanding the computational complexity of learning a generative model that is "indistinguishable" from the data distribution with respect to interesting metrics, such as integral probability metrics induced by various function classes, is an exciting future direction.
>
> **References**
>
> - [CKS24] Sitan Chen, Vasilis Kontonis, Kulin Shah. Learning general Gaussian mixtures with efficient score matching. arXiv preprint. 2024
> - [SCK23] Kulin Shah, Sitan Chen, Adam Klivans. Learning Mixtures of Gaussians Using the DDPM Objective. *NeurIPS* 2023.
> - [TV18] Yan Shuo Tan, Roman Vershynin. Polynomial Time and Sample Complexity for Non-Gaussian Component Analysis: Spectral Methods. *COLT* 2018.
> - [GS19] Navin Goyal, Abhishek Shetty. Non-Gaussian Component Analysis using Entropy Methods. *STOC* 2019.

---

> > ### Comment · Reviewer_q3zX · 2024-08-09
> >
> > Thank you to the authors for providing such a comprehensive response to my questions! This was an interesting foray into literature that I was not previously exposed to, and I appreciate the authors for engaging. I remain positive in my evaluation of the work, and best of luck to the authors. Thank you for providing an interesting read.

---

### Official Review · Reviewer_27f3 · 2024-07-12

**Soundness:** 3
**Presentation:** 3
**Contribution:** 3
**Rating:** 7
**Confidence:** 3

**Summary:**

Without knowing data distribution, it is computationally hard to estimate score function from data samples, such as the reverse step of diffusion models. One previous work shows that L^2-accurate score estimation along the forward process can help efficiently sampling from arbitrary data distribution. However, this works shows L^2-accurate score estimation is still computationally hard even with polynomial sample complexity. Finding a gap between statistical perspective and computational perspective leads to a set of hard distributions, which are “Gaussian pancakes” distributions. This works conclude computationally efficient  L^2-accurate score estimation should rely on stronger assumptions.

**Strengths:**

1. This work brings a set of solid future directions on the grounds of diffusion and computational complexity theory. It can guide fruitfully interesting research topics.
2. This work also bridges the score estimation with cryptography on computationally  indistinguishability, which is a important property in the cryptography. Also, this work interprets the hardness from lattice-based cryptographic perspective.

**Weaknesses:**

This work is lack of empirical evidences, especially those related to diffusion models.

**Questions:**

Can you remove one “between” in Line 32?

**Limitations:**

None limitation has been detected.

---

> ### Author Rebuttal · Authors · 2024-07-31
>
> We thank the reviewer for the positive feedback and rating. We are grateful for the reviewer’s appreciation of the significance of our main result as a bridge between score estimation and the cryptographic notion of computational indistinguishability, as well as the future research directions we have proposed in light of our findings. We also appreciate the reviewer pointing out the typo in our submission; it will be corrected in the revised version.
>
> We respectfully disagree with the view that lack of empirical evidence constitutes a weakness in our paper. While empirical evidence is indeed valuable in many areas of research, no amount of empirical evidence can establish computational hardness. Even if a large class of algorithms fails to solve a given problem, there is no guarantee that a different algorithm will not succeed. Instead, computer scientists and cryptographers begin with a few core hardness assumptions and rely on mathematical proofs to provide guarantees about computational hardness.
>
> Given this context, our work demonstrates such fundamental limits through theoretical analysis. Specifically, our paper provides mathematical proofs that establish the hardness of L2-accurate score estimation under standard cryptographic assumptions.
>
> Please let us know if there are any additional comments or questions. Thank you again for your time and consideration.

---

### Decision · Program_Chairs · 2024-09-25

**Decision:**

Accept (poster)

**Comment:**

The paper shows a statistical to computational gap for L2 score estimation under standard cryptographic assumptions. Reviewers agree this is an interesting and valuable contribution to understanding the computational complexity of problems related to diffusion models